# Sustained treatment of retinal vascular diseases with self-aggregating sunitinib microparticles

Hiroki Tsujinaka[1,4], Jie Fu[1,4], Jikui Shen[1,4], Yun Yu[2,4], Zibran Hafiz[1], Joshua Kays[2], David McKenzie [2], Delia Cardona[2], David Culp[3], Ward Peterson[2], Brian C. Gilger[3], Christopher S. Crean[2], Jin-Zhong Zhang[2], Yogita Kanan [1], Weiling Yu[2], Jeffrey L. Cleland[2], Ming Yang[2]*, Justin Hanes[1]* & Peter A. Campochiaro [1]*

Neovascular age-related macular degeneration and diabetic retinopathy are prevalent causes of vision loss requiring frequent intravitreous injections of VEGF-neutralizing proteins, and under-treatment is common and problematic. Here we report incorporation of sunitinib, a tyrosine kinase inhibitor that blocks VEGF receptors, into a non-inflammatory biodegradable polymer to generate sunitinib microparticles specially formulated to self-aggregate into a depot. A single intravitreous injection of sunitinib microparticles potently suppresses choroidal neovascularization in mice for six months and in another model, blocks VEGF-induced leukostasis and retinal nonperfusion, which are associated with diabetic retinopathy progression. After intravitreous injection in rabbits, sunitinib microparticles self-aggregate into a depot that remains localized and maintains therapeutic levels of sunitinib in retinal pigmented epithelium/choroid and retina for more than six months. There is no intraocular inflammation or retinal toxicity. Intravitreous injection of sunitinib microparticles provides a promising approach to achieve sustained suppression of VEGF signaling and improve outcomes in patients with retinal vascular diseases.

[1] The Center for Nanomedicine, The Wilmer Eye Institute, Johns Hopkins University School of Medicine, Baltimore, MD, USA. [2] Graybug Vision, Inc., Redwood City, CA, USA. [3] Powered Research, LLC, Research Triangle Park, Triangle Park, NC, USA. [4]These authors contributed equally: Hiroki Tsujinaka, Jie Fu, Jikui Shen, Yun Yu. *email: myang@graybug.com; hanes@jhmi.edu; pcampo@jhmi.edu

Age-related macular degeneration (AMD) is a highly prevalent cause of vision loss[1]. It is a neurodegenerative disease characterized by deposits called drusen, diffuse thickening of Bruch's membrane beneath the retinal pigmented epithelium (RPE), and slow death of photoreceptors, RPE, and choriocapillaris beneath the macula resulting in gradual loss of central vision. A subgroup of AMD patients develop choroidal neovascularization (NV) in which new vessels originate in the choroid and extend under the RPE (type 1 choroidal NV) or through the RPE into the subretinal space (type 2), or new vessels originate from the deep capillary bed of the retina and grow through the outer nuclear layer (ONL) containing the nuclei of the photoreceptors into the subretinal space (type 3). Patients with any of these types of choroidal NV are said to have neovascular AMD (NVAMD) and often experience rapid reduction in vision due to leakage of plasma from NV causing fluid to collect under or within the macula, but the distinction is important because patients with type 3 choroidal NV carry a high risk of retinal atrophy, another source of visual loss[2]. The risk of developing NVAMD increases with advancing age and the severity of drusen and/or pigmentary changes in the macula; in the Age Related Eye Disease Study, the risk of developing NVAMD within 10 years was 48% for the oldest participants with the most extensive drusen and pigmentary changes[3]. Once NVAMD develops in one eye, there is a high risk for its development in the fellow eye.

Studies in models of type 2 or type 3 choroidal NV like that seen in NVAMD implicated vascular endothelial growth factor (VEGF) as a critical stimulator[4–6]. Clinical trials showed that monthly intravitreous injections of the VEGF antagonist ranibizumab provided substantial improvement and stabilization of vision in patients with NVAMD[7,8]. However, in an extension study in which visits and injections were only required every 3 months, most of the visual gains obtained during 1 year of frequent injections were lost within a year[9]. A large multicenter clinical trial comparing monthly injections of ranibizumab or bevacizumab to injections only when intraretinal or subretinal fluid was present showed only a small decrease in overall number of injections in the latter groups over the course of 2 years and mean improvements in visual acuity were significantly better in patients receiving monthly injections[10]. Patients were returned to standard care and over the next 3 years the mean number of injections per year was reduced and the mean best-corrected visual acuity in each of the 4 groups declined to a level worse than baseline (loss of 3.3 letters from baseline)[11]. In contrast, a study in which patients with NVAMD were treated aggressively and given an average of 10.4 injections per year for 5 years, there was a large mean improvement of 14.1 letters[12]. In clinical practice, the mean number of intravitreous injections of a VEGF-neutralizing protein per year is substantially less than that in clinical trials and visual outcomes are much worse[13]. These data indicate that sustained suppression of VEGF provides the best visual outcomes in patients with NVAMD.

Diabetic retinopathy and retinal vein occlusion are ischemic retinopathies that are prevalent causes of vision loss in working age individuals[14,15]. The most common cause of vision loss in each is macular edema due to leakage from retinal blood vessels[16–18]. Pilot trials demonstrated that intraocular injections of ranibizumab provided substantial reduction of intraretinal fluid in the macula and improvement in vision[19–21]. These findings were confirmed in phase 3 studies that led to FDA approval of ranibizumab for treatment of diabetic macular edema and retinal vein occlusion[22–24]. Follow-up studies showed that suppression of VEGF reduced progression of retinal vessel closure and caused improvement in perfusion in some patients with retinal vein occlusion or diabetic macular edema (DME)[25,26], and improved

background diabetic retinopathy in patients with DME[27]. These two effects of VEGF suppression are related because slowly progressive closure of retinal vessels is strongly associated with progression of background diabetic retinopathy[28]. Thus, increased VEGF in the retina drives disease progression of diabetic retinopathy and retinal vein occlusion and sustained suppression of VEGF can halt progression and cause improvement.

We have developed an approach through which sustained suppression of VEGF signaling is achieved. We have incorporated sunitinib, a small molecule inhibitor of VEGF receptors and other tyrosine kinases that has been approved as an oral agent for treatment of renal cell carcinoma[29,30], into microparticles (MPs) composed of blends of poly(lactic-co-glycolic acid) (PLGA) and PLGA conjugated to polyethylene glycol (PLGA-PEG). A different polymer MP formulation and manufacturing process is key to achieving three critical goals: (1) sustained release of effective levels of sunitinib for many months, (2) elimination of ocular inflammation typically observed following intravitreous injection of PLGA MPs, and (3) intraocular injection triggered self-aggregation of MPs into a depot near the site of injection that reduces dispersion of MPs into the visual axis. Here, we report the efficacy and pharmacokinetics of sunitinib MPs after intravitreous or subconjunctival injection in animal models of retinal and choroidal vascular diseases.

## Results

**Formulation and characterization of sunitinib MPs.** Our previous work showed that a dense PEG coating on polymer MPs is critical for minimizing the inflammation associated with intraocular injection of MPs without a PEG coating[31] therefore, sunitinib MPs were produced using a blend of PLGA-PEG and PLGA polymers to improve biocompatibility with ocular tissues. We next sought to improve the drug loading of sunitinib MPs to ensure that an adequate amount of sunitinib is delivered to provide therapeutic levels for several months by a single intravitreous injection with volume ≤50 μl. Because sunitinib has high solubility in acidic aqueous solutions (pH 1.2–6.8) and the solubility rapidly decreases at pH greater than 6.8, a relatively large volume of aqueous solution of pH 7.4 was used as the continuous phase, into which an organic solution containing dissolved sunitinib and polymers was emulsified to produce sunitinib MPs[32]. Due to the low aqueous solubility of sunitinib at pH 7.4, drug loss from the organic phase during MP production was minimized, thus resulting in high drug loading of over 10% (w/w) in MPs[33]. As shown in Fig. 1a, sunitinib was released from MPs in a sustained manner for over 2 months with minimal burst under sink conditions in vitro. The MPs designed for mouse experiments were smaller ($13 \pm 6$ μm) than those designed for rabbit and minipig experiments ($32 \pm 9$ μm), and they had a slightly faster release profile, but in the same range (Fig. 1b, closed squares). In contrast to sunitinib MPs, sunitinib drug suspension (Fig. 1b, open squares) was rapidly eluted and dissolved in release medium in vitro. The drug loading of the sunitinib MPs used in mice and rabbits was 10.6% and 12.0% (w/w), respectively.

Dispersion of MPs throughout the eye can block the visual axis and interfere with vision; therefore, MPs were modified to promote aggregation following intravitreous injection. As shown in Fig. 1c, following an injection of 50 μl sunitinib MP suspension and incubation at 37 °C for 2 h, a solid depot formed in a hyaluronate solution that could be grasped and isolated with forceps (Fig. 1c). One day after intravitreous injection of 50 μl of sunitinib MP suspension though a 27 g needle into the inferior vitreous of a rabbit, the aggregate of MPs could not be seen within the eye in primary position with a 30° fundus camera; however, with a contact lens on the eye which is grasped with a forceps and

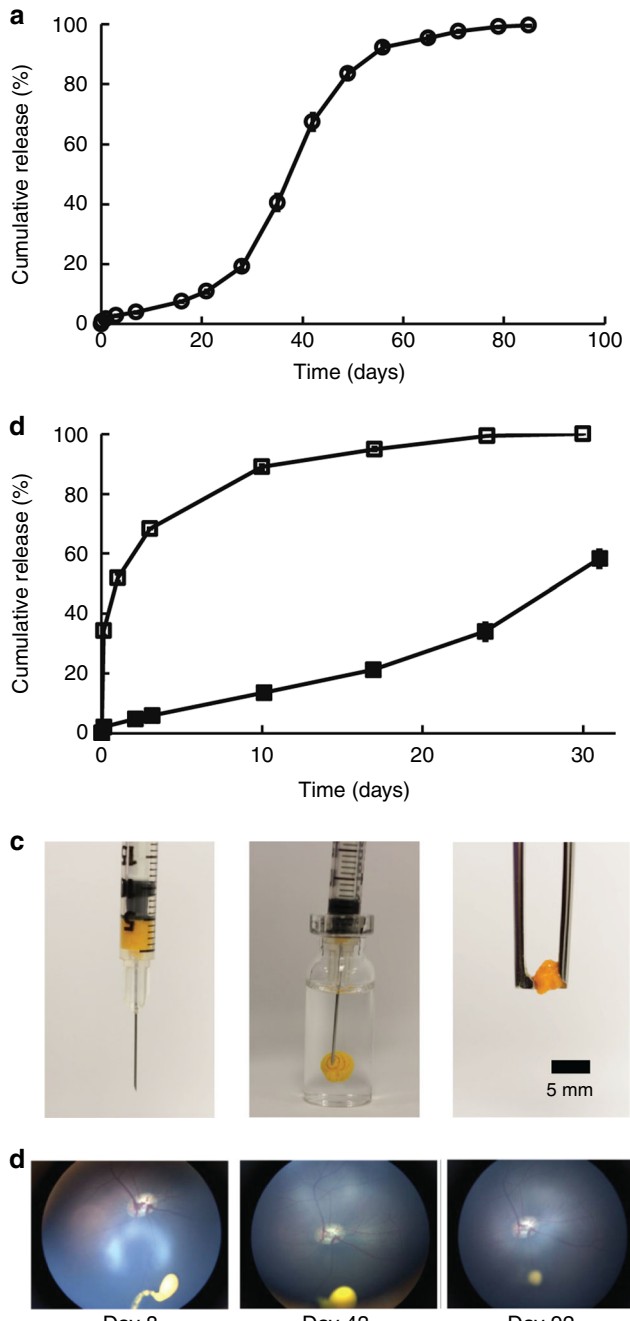

**Fig. 1 Characterization of sunitinib MPs and degradation of sunitinib MP depot in the vitreous of minipigs. a** Sunitinib MPs used in rabbit and minipig experiments release drug in a sustained manner for nearly 3 months in vitro at 37 °C under sink conditions. Circles are the means for two independent experiments. **b** Sunitinib drug suspension (open squares) was rapidly eluted and dissolved in release medium in vitro, while MPs containing an equivalent amount of sunitinib (closed squares) achieved a much more sustained release over time. **c** An injection of 50 μl sunitinib MP suspension in a hyaluronate solution at 37 °C leads to the formation of a solid depot that can be isolated using a pair of forceps. **d** Wide-field fundus photographs showing the reduction of depot size over time (Day 8, 43, and 92 post dose) in vitreous after an intravitreous injection of MP suspension containing 1 mg sunitinib in minipigs. The yellow spot near the bottom of each photo indicates the sunitinib MP depot. Source data are provided as a Source Data file.

turned inferiorly, a discrete yellow aggregate was seen (Supplementary Movie 1). Similarly, following an intravitreous injection of 50 μl of sunitinib MPs in minipigs, the depot was only visible in the inferior vitreous by using wide-field fundus photography. As shown in Fig. 1d, the size of the depot gradually decreased as the MPs degraded and released sunitinib over time.

**Sunitinib MPs suppress type 2 murine choroidal NV.** The murine model of laser-induced rupture of Bruch's membrane results in choroidal NV that is similar to type 2 choroidal NV in patients with NVAMD, because the new vessels originate from the choroid and penetrate through Bruch's membrane and the RPE into the subretinal space[34]. Studies in this model helped to implicate VEGF as a critical stimulus for NVAMD[4] and predicted the clinical benefits seen with aflibercept, a recombinant VEGF-neutralizing protein commonly used to treat patients with NVAMD[6,35]. In order to assess the efficacy of sunitinib MPs over time, C57BL/6 mice were given an intravitreous injection of MPs containing 10 or 1 μg sunitinib in one eye and empty MPs in the fellow eye and then had laser-induced rupture of Bruch's membrane at 3 locations in each eye at time points ranging from 1 to 24 weeks after injection. Compared with empty MP fellow eye controls, the mean area of choroidal NV at Bruch's membrane ruptures sites was significantly less in eyes injected with MPs containing 10 μg sunitinib at each time point through week 24 (Fig. 2a). The mean area of choroidal NV at Bruch's membrane rupture sites/eye was significantly less in eyes injected with MPs containing 1 μg sunitinib compared with fellow eye controls at 9 and 15 weeks after injection, but not 20 or 24 weeks after injection (Fig. 2b). One of 81 eyes injected with sunitinib MPs in Fig. 2a, b had mild cataract, and the fellow eye of that mouse that was injected with empty MPs also had mild cataract. In contrast, intravitreous injection of doses of free sunitinib >0.5 μg caused severe cataract and there was no significant difference in mean area of choroidal NV in eyes injected with 0.5 or 0.1 μg of free sunitinib compared with fellow eyes injected with phosphate-buffered saline (PBS) (Fig. 2c). Thus, intravitreous injection of nontoxic doses of free sunitinib had no efficacy a week after injection. There was significant suppression of choroidal NV 1 week after injection of 40 μg of aflibercept, but not 9 or 15 weeks after injection (Fig. 2d).

**Subconjunctival sunitinib MPs suppress type 2 choroidal NV.** Subconjunctival injection is painless and allows injections of larger volumes. Mice had a 2 μl subconjunctival injection of MPs containing 20 or 2 μg sunitinib or empty MPs in one eye and no injection in the fellow eye. After 1 week, Bruch's membrane was ruptured in 3 locations in each eye; the area of choroidal NV at Bruch's membrane rupture sites was significantly less in eyes injected with MPs containing 20 or 2 μg sunitinib, compared with corresponding fellow eyes or eyes injected with empty MPs (Fig. 3). There was no significant difference in mean area of choroidal NV at sites of Bruch's membrane rupture in uninjected fellow eyes in the sunitinib MP-injected groups vs. the empty MP-injected groups, indicating that the effect after subconjunctival injection of sunitinib MPs was local and not due to systemic exposure to sunitinib.

**Sunitinib MPs suppress murine type 3 choroidal NV.** Approximately, 30% of patients with NVAMD have type 3 choroidal NV (previously called retinal angiomatous proliferation) in which new vessels originate from the deep capillary bed of the retina, grow through the photoreceptor layer, and form networks of NV in the subretinal space[36]. Transgenic mice in which the

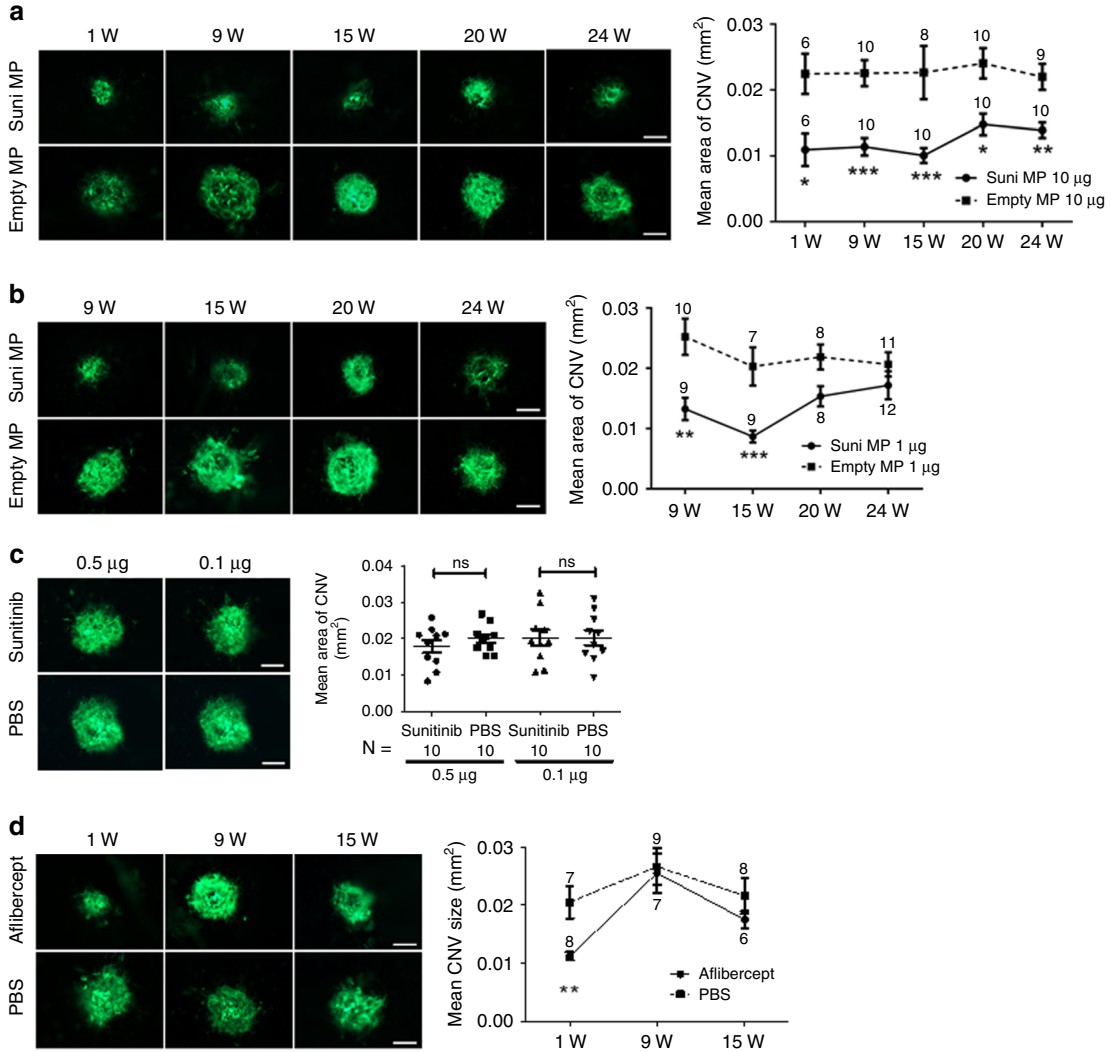

**Fig. 2 Effect of sunitinib microparticles in a mouse model of type 2 choroidal neovascularization.** C57BL/6 mice received a 1 µl intravitreous injection of sunitinib (Suni) microparticles (MP) containing 10 or 1 µg of sunitinib in one eye and an equivalent amount of empty MP in the fellow eye, or they received an injection of 0.5 or 0.1 µg of free sunitinib in one eye and PBS in the fellow eye, or 40 µg of aflibercept in one eye and PBS in the fellow eye. At various time points after injection ranging from 1 to 24 weeks, the mice had rupture of Bruch's membrane by laser photocoagulation at 3 locations in each eye. One week after laser, mice were euthanized and choroidal flat mounts were stained with FITC-labeled Griffonia Simplicifolia Agglutinin (GSA) lectin and the area of choroidal neovascularization (CNV) was measured at Bruch's membrane rupture sites by an investigator masked with regard to treatment. The three values from each eye were averaged to give a single experimental value. The mean (±SEM) area of CNV was significantly less in eyes injected with MP containing 10 µg sunitinib compared with fellow eyes injected with empty MP at each time point through 24 weeks (**a**). Eyes injected with MP containing 1 µg sunitinib showed a significant reduction in mean area of CNV compared with corresponding controls at 9 and 15 weeks, but not 20 and 24 weeks (**b**). Eyes injected with 0.5 or 0.1 µg of free sunitinib showed no significant reduction in area of CNV compared to fellow eyes injected with PBS at 1 week (**c**). Eyes injected with 40 µg of aflibercept showed a significant reduction in mean CNV area compared with fellow eye controls at 1 week, but not 9 or 15 weeks (**d**). Number of animals used in each group are shown in the graph or below the graph. *$p < 0.05$; **$p < 0.01$, ***$p < 0.001$ by Mann–Whitney compared with fellow eye control at that time point. Bar = 100 µm. Source data are provided as a Source Data file.

*rhodopsin* promoter drives expression of VEGF in photoreceptors (*rho/VEGF* mice) provide a model of type 3 choroidal NV[37,38], which along with mice with type 2 choroidal NV was used to first demonstrate the efficacy of aflibercept[6]. At postnatal day (P) 14, one eye of *rho/VEGF* mice was injected with MPs containing 10 µg sunitinib and the fellow eye was injected with an equivalent mass of empty MPs and, at weekly intervals, mice were euthanized and retinas were stained with *Griffonia simplicifolia* agglutinin (GSA) lectin and flat mounted with the photoreceptor side up showing dark green tufts of NV on the outer surface of the retina (sub-retinal space). Some of the tufts are surrounded by RPE (black pigment) and feeder vessels, portions of which are out-of-focus,

because they extend from the deep capillary bed to the plane of focus, the outer surface of the retina. Compared with eyes injected with empty MPs, fellow eyes injected with MPs containing 10 µg sunitinib showed fewer tufts of subretinal NV by visual inspection and significantly less mean area of NV per retina at P21, P28, P35, and P42 (Fig. 4a). Eyes injected with 40 µg of aflibercept had fewer buds of NV and significantly less mean area of NV per retina than control fellow eyes injected with PBS at P21, but not P28, P35, or P42 (Fig. 4b).

In clinical care, anti-VEGF agents are injected in eyes with established choroidal NV, and their major effect is to reduce leakage from the NV thereby reducing fluid in the retina, and

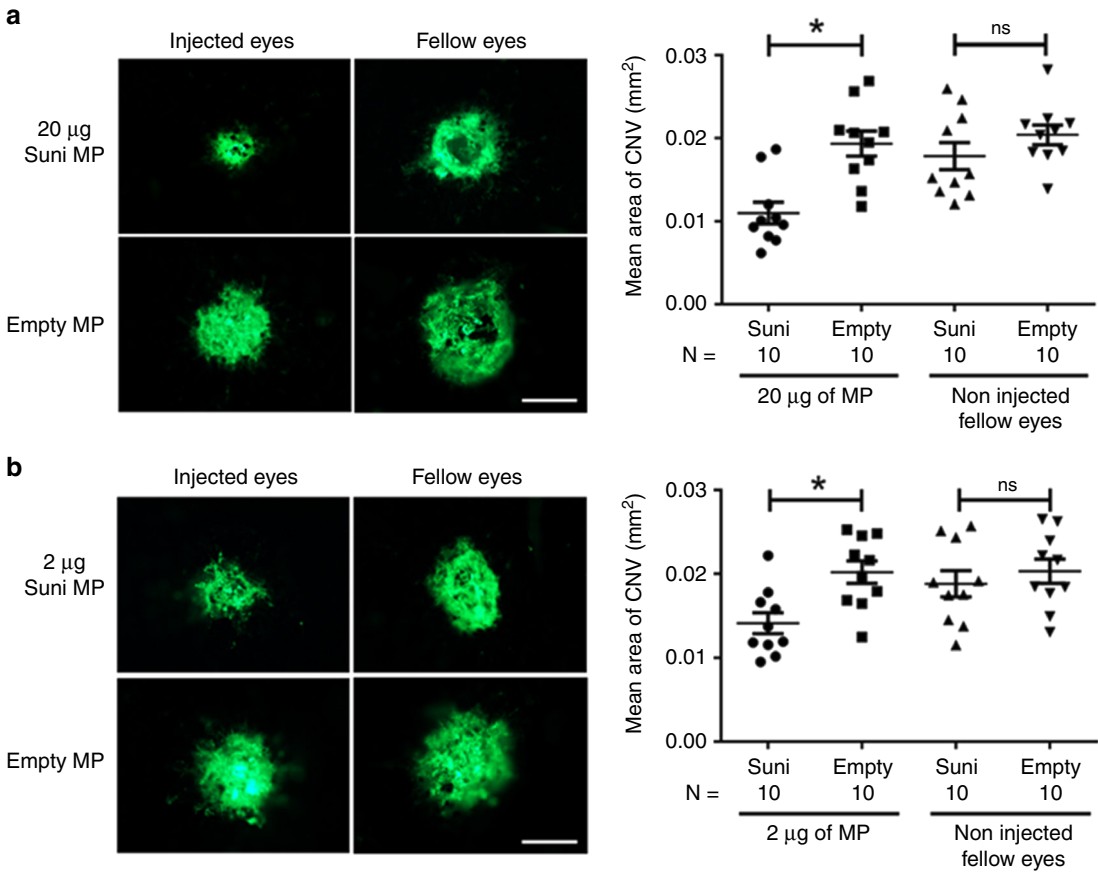

**Fig. 3 Subconjunctival injection of sunitinib MPs suppresses type 2 choroidal NV.** C57BL/6 mice received a 2 μl subconjunctival injection of microparticles (MP) containing 20 or 2 μg of sunitinib (Suni) or empty MP in one eye and no injection in the fellow eye. At 1 week after injection, Bruch's membrane was ruptured by laser photocoagulation at three locations in each eye. One week after laser, mice were euthanized and choroidal flat mounts were stained with FITC-labeled GSA lectin and the area of choroidal neovascularization (CNV) was measured at Bruch's membrane rupture sites by an investigator masked with regard to treatment. The three values from each eye were averaged to give a single experimental value. The mean (±SEM) area of CNV was significantly less in eyes injected with MP containing 20 μg (**a**) or 2 μg (**b**) sunitinib vs. those injected with empty MP, but there was no significant difference between fellow eyes for each group. *$p < 0.05$ for difference from all other groups by Kruskal Wallis test followed by Dunn's test. Bar = 100 μm. Source data are provided as a Source Data file.

improving vision. Fluorescein angiograms of *rho/VEGF* mice at P28, when there is extensive subretinal NV, does not have sufficient resolution to show individual buds of NV, but shows collections of extravascular fluid scattered throughout the retina, particularly in the posterior pole (Fig. 4c, top row). One week after intravitreous injection of MPs containing 10 μg sunitinib or an equivalent mass of empty MPs, sunitinib MP-injected eyes showed a marked reduction in extravascular fluorescein, while empty MP-injected eyes did not (Fig. 4c, bottom row). Fluorescein angiography provides qualitative assessment of leakage that is relevant to what is done in clinical practice, but it is not quantitative; however, measurement of serum albumin in the vitreous provides a precise, objective measure of vascular leakage[39]. Compared to eyes injected with empty MPs, those injected with MPs containing 10 μg sunitinib had significantly less mean concentration of albumin in the vitreous (Fig. 4c, right). Eyes injected with 40 μg of aflibercept showed significant reduction in mean vitreous albumin concentration compared with those injected with PBS. Measurement of the amount of Evans Blue dye that leaks into the retina after systemic injection provides another assessment of retinal vascular leakage. There was significantly less Evans Blue dye in the retina of eyes injected with MP containing 10 μg sunitinib compared with those injected with empty MP (Fig. 4e).

**Sunitinib MPs reduce photoreceptor death in eyes with type 3 choroidal NV.** The incidence of macular atrophy is high in patients with type 3 choroidal NV treated with anti-VEGF agents[2]. Measurement of ONL which contains the nuclei of photoreceptors provides an assessment of photoreceptor survival commonly used in models of inherited retinal degeneration. Rhodopsin kinase is an enzyme involved in phototransduction and its levels in retinal homogenates correlates with the number of healthy photoreceptors. In *rho/VEGF* mice with type 3 choroidal NV, photoreceptor cell death is accompanied by thinning of the ONL and reduction in retinal levels of rhodopsin kinase, which are substantially prevented by the potent antioxidant N-acetylcysteine, but are neither exacerbated nor improved by VEGF suppression[40]. This indicates that oxidative damage is a major contributor to photoreceptor cell death that occurs in eyes with type 3 choroidal NV. At P14, *rho/VEGF* mice were given an intravitreous injection of MPs containing 10 μg sunitinib in one eye and empty microparticles in the fellow eye or 40 μg of aflibercept in one eye and PBS in the fellow eye. At P42, mean ONL thickness was significantly greater in sunitinib MP-injected eyes compared with empty MP-injected eyes at 3 of 6 measurement locations along the vertical meridian of the retina through the optic nerve, but there was no significant difference between aflibercept-injected and PBS-injected eyes (Fig. 5a, b). At P49,

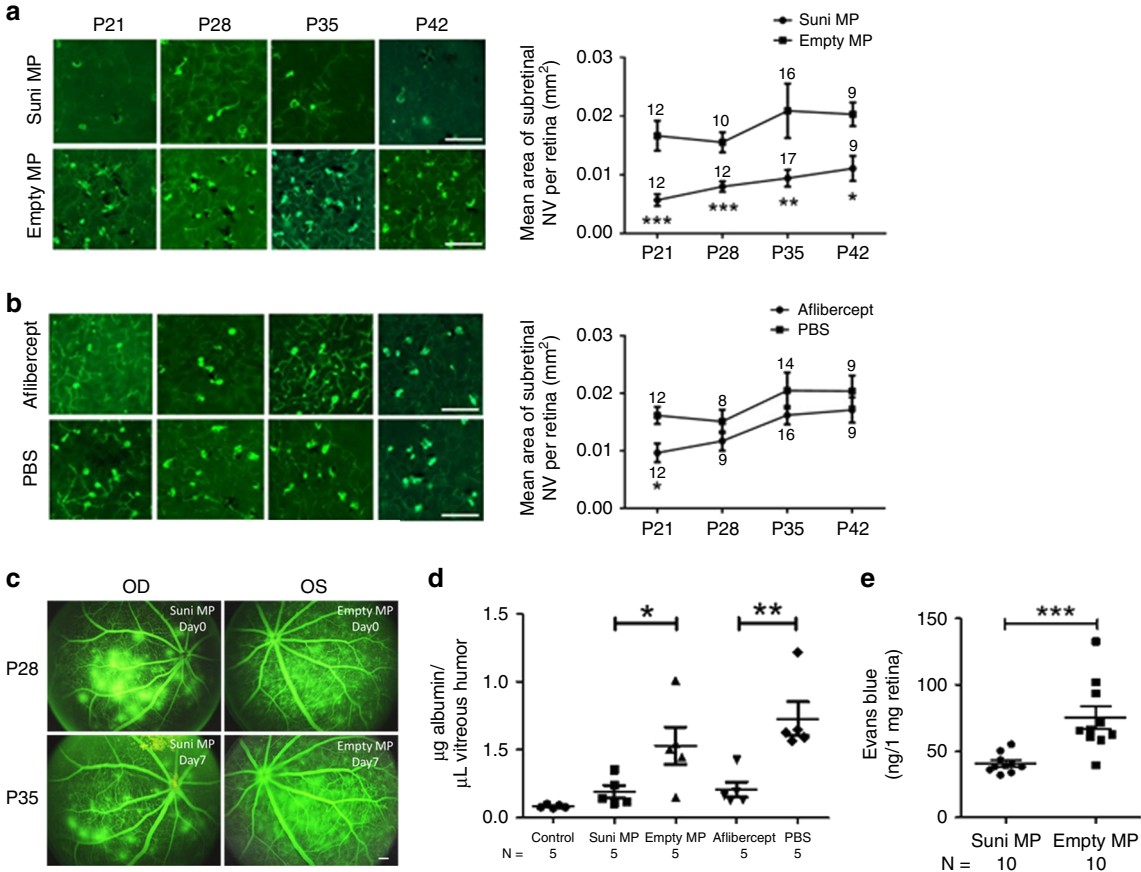

**Fig. 4 Sunitinib MPs suppress murine type 3 choroidal NV substantially longer than aflibercept.** At P14, *rho/VEGF* mice had intravitreous injection of MP containing 10 μg sunitinib (Suni MP) in one eye and empty MPs in the other eye or 40 μg of aflibercept in one eye and PBS in the other eye. At P21, P28, P35, or P42, mice were euthanized and retinal flat mounts were stained with FITC-labeled GSA lectin. The total area of subretinal NV per retina was measured by image analysis. Compared with empty MP-injected eyes, those injected with Suni MPs had significantly lower mean (±SEM) area of NV per retina at each time point (**a**). Compared with PBS-injected eyes, those injected with aflibercept had significantly lower mean (±SEM) area of NV per retina at P21, but not P28, P35, or P42 (**b**). At P28, *rho/VEGF* mice had fluorescein angiography showing severe leakage resulting in large collections of extravascular fluorescein (**c** top row). One eye was injected with MP containing 10 μg of sunitinib or 40 μg of aflibercept and the other with empty MPs or PBS and after 1 week repeat fluorescein angiography showed less leakage in sunitinib MP-injected eyes, but not empty MP-injected eyes (**c** bottom row, scale bar = 100 μm). Vitreous samples were obtained and the mean (±SEM) concentration of vitreous albumin measured by ELISA was significantly less in Suni MP-injected eyes vs. empty MP-injected eyes and similar to untreated control eyes (**d**). Mean vitreous albumin was also significantly less in aflibercept-injected eyes vs. PBS-injected eyes (**d**). The experiment described in (**c**) and (**d**) was repeated in P28 *rho/VEGF* mice using a different outcome measure, leakage of intravascular Evans Blue dye into the retina as described in Methods. **e** The mean (±SEM) concentration of Evans Blue dye in the retina was significantly less in eyes injected with sunitinib MP compared with those injected with empty MP. Number of animals used in each group are shown in the graph or below the graph. *$p < 0.05$; **$p < 0.01$; ***$p < 0.001$ by Mann Whitney from corresponding control. Bar = 100 μm. Source data are provided as a Source Data file.

immunoblots of retinal homogenates from sunitinib MP-injected eyes had significantly greater rhodopsin kinase/actin ratio compared with those from empty MP-injected eyes, but there was no difference between aflibercept- and PBS-injected eyes (Fig. 5c, d).

**Sunitinib MPs reduce VEGF-induced leukostasis/ nonperfusion.** VEGF plays a critical role in the pathogenesis of diabetic retinopathy as well as NVAMD. Increased levels of VEGF in the retina cause leukostasis and closure of retinal vessels exacerbating retinal hypoxia[41], which is strongly associated with progression of diabetic retinopathy[28,42]. Adult C57BL/6 mice were given an intravitreous injection of 200 ng of VEGF in each eye one week after injection of MPs containing 10 μg sunitinib or 40 μg of aflibercept in one eye and an equivalent mass of empty MPs or PBS in the fellow eye. Twenty-four hour later, mice were perfused with PBS through the left ventricle to remove all non-adherent erythrocytes and leukocytes, and then perfused with conconavalin

A to stain remaining adherent leukocytes. Compared with eyes injected with empty MPs, those injected with sunitinib MPs had a significant reduction in the mean number of adherent intravascular leukocytes (Fig. 6a). Similarly aflibercept-injected eyes had a significant reduction in the mean number of adherent intravascular leukocytes compared with PBS-injected eyes. This experiment was repeated, but the number of adherent leukocytes in vessels was visualized by immunohistochemical staining with anti-CD45 antibody and this also showed a significant reduction in leukostasis in eyes injected with sunitinib MPs compared with empty MPs (Fig. 6b).

The constant moderate expression of VEGF in *rho/VEGF* mice not only causes type 3 choroidal NV in the outer retina, but also causes gradual onset of leukostasis and retinal vessel closure in the inner retina thereby mimicking these aspects of diabetic retinopathy[41]. *Rho/VEGF* mice (≥3 months old) had fluorescein angiography which showed areas of retinal nonperfusion and

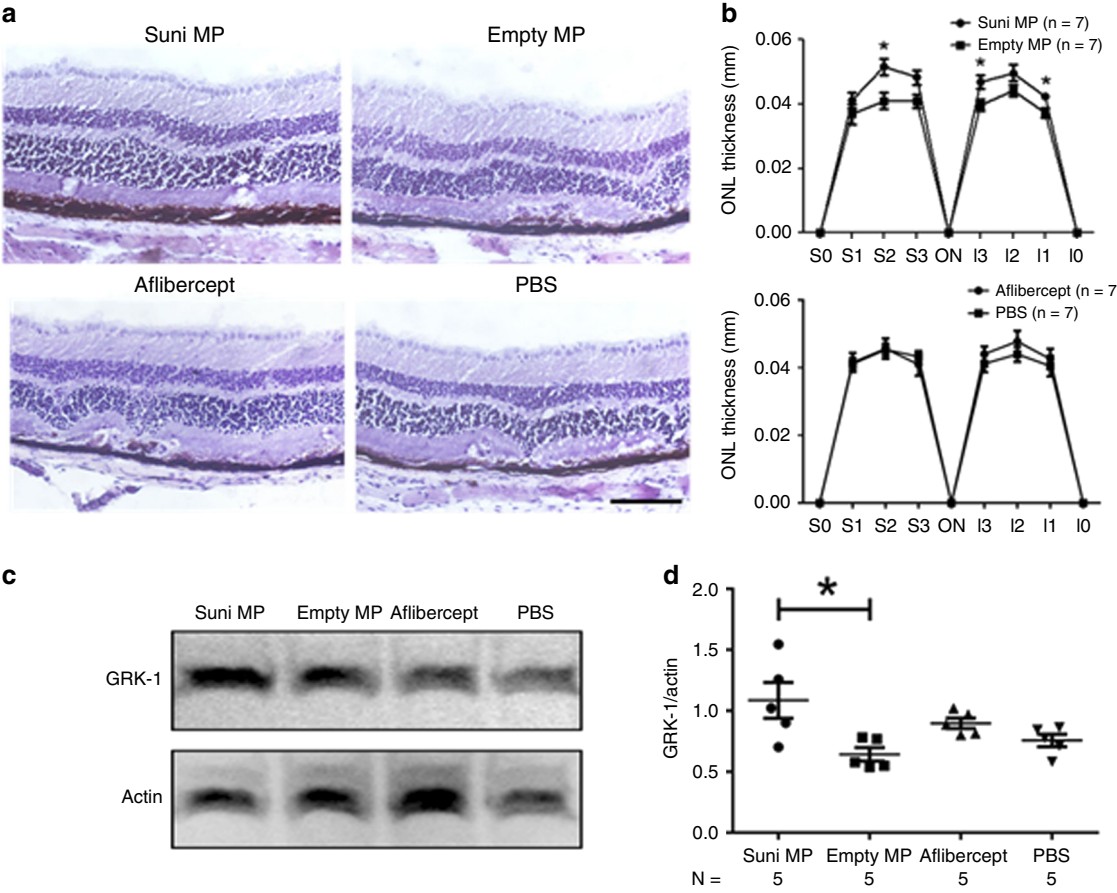

**Fig. 5 Intravitreous injection of sunitinib MPs in mice with type 3 CNV reduces photoreceptor cell death.** *Rho/VEGF* mice were given an intravitreous injection of 10 μg of sunitinib microparticles (Suni MP) in one eye and 10 μg of empty MP in the other eye or 40 μg of aflibercept in one eye and PBS in the fellow eye. At P42, mice ($n = 7$ for each group) were euthanized and serial frozen ocular sections were cut from the superior pole of the eye (S0) to the inferior pole (I0) and sections 25% (S1 and I1), 50% (S2 and I2), and 75% (S3 and I3) of the distance between each pole and the optic nerve (ON) were stained with hematoxylin and outer nuclear layer (ONL) thickness was measured by image analysis by a masked investigator. The ONL of sections from the S2 location of Suni MP-injected eyes appeared thicker than those from empty MP-injected eyes, but those from aflibercept-injected eyes appeared similar to those from PBS-injected eyes (**a** scale bar = 100 μm). The mean (±SEM) ONL thickness was significantly greater at three of six locations in Suni MP-injected eyes compared with empty MP-injected eyes, but there was no difference between aflibercept- and PBS-injected eyes (**b**). At P49, mice ($n = 5$ for each group) were euthanized and immunoblots of retinal homogenates from Suni MP-injected eyes showed prominent bands for rhodopsin kinase (GRK-1) (**c**). Denistometry showed that the mean (±SEM) GRK-1/Actin ratio was significantly greater in retinas from Suni MP-injected eyes compared with empty MP-injected fellow eyes, but not aflibercept-injected eyes vs. PBS-injected fellow eyes (**d**). *$p < 0.05$ by Mann Whitney for difference from fellow eye empty MP control. Source data are provided as a Source Data file.

then had intravitreous injection of MPs containing 10 μg sunitinib or an equivalent mass of empty MPs. Repeat fluorescein angiography 2 weeks after injection showed reperfusion of some previously nonperfused areas in sunitinib MP-injected retinas (Fig. 6c).

**Pharmacokinetics of sunitinib MPs in mice and rabbits**. As shown in Fig. 7a, at 30 days after an intravitreous injection of sunitinib MPs in C57BL/6 mice, 52% of the initial sunitinib MP dose still remained in the eye. In comparison, only 9% of the initial dose still remained in the eye 30 days after injection of an equivalent amount of sunitinib drug suspension. The data also correlate well with the in vitro drug release profiles in Fig. 1b. The results suggest that when not encapsulated in polymer MPs, sunitinib was rapidly cleared from the eye. In contrast, a much more sustained release of drug was achieved from sunitinib MPs.

A single 50 μl intravitreous injection of MP suspension containing 1 mg sunitinib or a single 200 μl subconjunctival injection of MP suspension containing 2 mg sunitinib was administered to pigmented New Zealand rabbits and the rabbits

were euthanized at selected time points to evaluate the levels of sunitinib in ocular tissues. Within 1 week after the intravitreous injection, the levels of sunitinib in the retina and RPE/choroid were approximately 100-fold greater than the reported level required to inhibit VEGF receptors (8 nM; ~4 ng/g)[43]. The levels in the retina and RPE/choroid continued to increase and peaked at 3 months (Fig. 7b). Analysis for remaining sunitinib content in vitreous indicated complete release of sunitinib from the depot in 3 months. However, therapeutic levels of sunitinib were detected in the RPE/choroid for more than 6 months. The extended drug exposure is likely due to binding of sunitinib to melanin granules in the RPE/choroid, which served as a secondary depot for sunitinib.

The capacity of melanin to bind sunitinib was characterized in vitro by co-incubating sunitinib (100 ng to 2 mg) in 10 mg melanin (purified from sepia officialis) in PBS and quantifying the amount of free sunitinib in solution after 1 h. Melanin-bound sunitinib was calculated by subtracting free sunitinib in solution from total sunitinib. As the sunitinib-to-melanin mass ratio increased from 0.001 to 20%, the percentage of melanin-bound

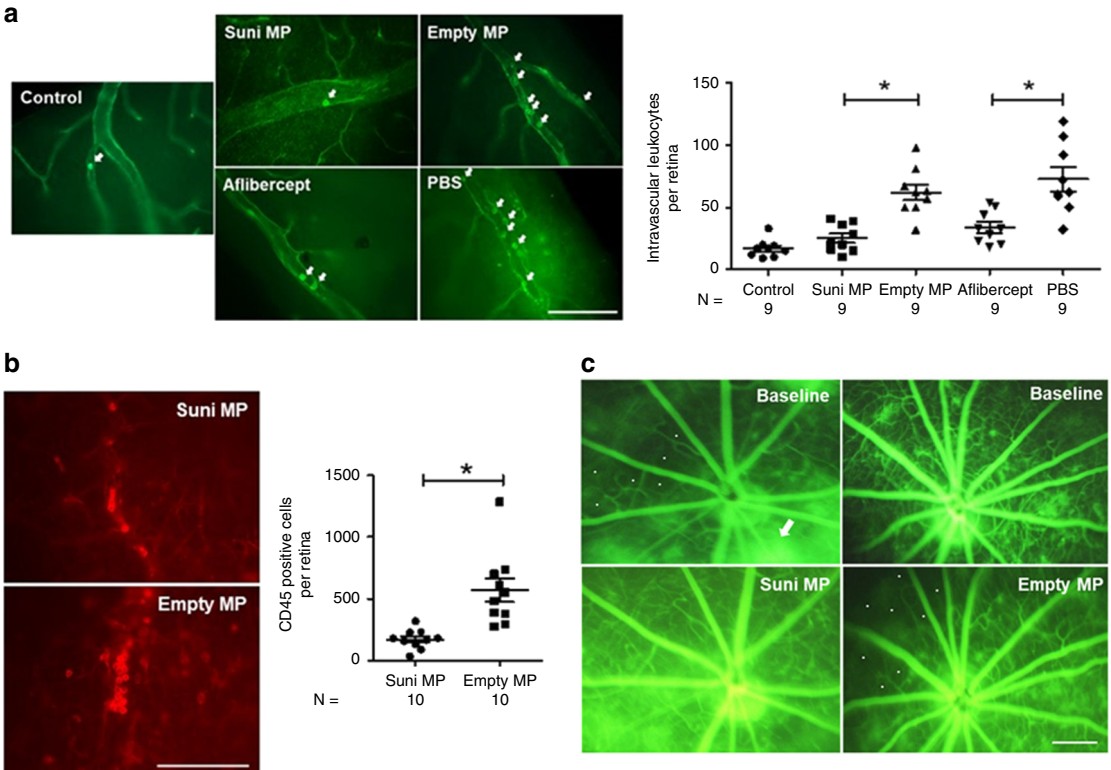

**Fig. 6 Sunitinib MP reduce leukostasis and improve retinal perfusion in rho/VEGF mice.** *Rho/VEGF* mice were given an intravitreous injection of 10 µg of Suni MP or 40 µg of aflibercept in one eye and 10 µg of Empty MP or PBS in the other eye. After 1 week, mice were perfused through the left ventricle with PBS to remove erythrocytes and leukocytes and then perfused with FITC-concanavalin A. Examination of retinal flat mounts by fluorescence microscopy showed a significant reduction in mean number of adherent intravascular leukocytes per retina in Suni MP-injected eyes compared with Empty MP-injected fellow eyes and in aflibercept-injected eyes compared with PBS-injected fellow eyes (**a** scale bar = 100 µm; *$p < 0.001$ by Mann Whitney for difference from corresponding control). The experiment was repeated but leukostasis was visualized by immunohistochemical staining with anti-CD45 which showed a significant reduction in leukocytes within retinal vessels in eyes injected with Suni MPs vs. those injected with empty MPs (**b** scale bar = 100 µm; *$p < 0.001$ by Mann Whitney for difference from corresponding fellow eye control). Rho/VEGF mice had fluorescein angiography at baseline and then were given an intravitreous injection of 10 µg of Suni MP in one eye and 10 µg of Empty MP in the other eye. Some areas of retinal nonperfusion at baseline showed improved perfusion 1 week after injection of Suni MP, but this was not seen in Empty MP-injected eyes (**c** scale bar = 250 µm). Source data are provided as a Source Data file.

sunitinib decreased from 99.8 to ~80%. Thus, at a ratio of <10, 96–99.8% of the drug is bound to melanin and binding saturation occurs when this ratio is 10–20%. The combination of sunitinib release from MPs and subsequent release from the melanin secondary reservoir likely accounts for the sustained drug level seen in the retina and RPE/choroid for more than 6 months after a single intravitreous injection. Melanin content in pigmented rabbit choroid is in the range of 20–120 ng melanin per µg tissue[44], which is similar to that in humans[45].

A similar pharmacokinetic profile was obtained via subconjunctival injection. Within 1 week after administration of MPs containing 2 mg sunitinib, the levels of sunitinib in the retina and RPE/choroid were approximately 200-fold greater than the reported level required to inhibit VEGF receptors. The levels in the retina and RPE/choroid continued to increase through Day 28, the last time point in this pilot study (Fig. 7c).

**Ocular tolerability and safety of sunitinib MPs in minipigs.** Following 2 repeat intravitreous injections 20 weeks apart of 0.25, 0.5, or 1 mg/eye, sunitinib MPs were well-tolerated during a 40-week observation period with regard to all endpoint assessments included in a repeat-dose ocular toxicity study in minipigs. The no-observed-adverse-effect-level is above the highest dose tested in this study. There were no sunitinib MP-associated ocular

findings with exception of a transient, expected minor, focal yellow discoloration to vitreous humor and lens observed in some animals. This discoloration resolved over time and was considered secondary to the presence of sunitinib. It was not considered adverse. Sporadic presence of pigment on anterior lens capsule during pretest and dosing period in the left (control PBS-injected) or the right (sunitinib MP-injected) eyes. This was considered incidental as it appeared sporadic and also during pretest and in the left control eyes.

Group mean average values for IOP readings in males were considered within normal limits at all time points with exception of Group 1 in both eyes at the Day 198 time point and Group 3 in the left eye at the Day 43 and Day 142 time points. These findings were considered incidental and not associated with test article administration since they affected control eyes. Group mean average values for IOP readings in females were considered within normal limits at all time points.

Electroretinography (ERG) assessments were also conducted in this study on all animals pretest and Weeks 19, 30, and 39 using the Roland Consult Retiport Gamma ERG system. Amplitude and latency values were measured from tracings. Each ERG assessment consisted of a series of stimuli with flash intensity of 0.0025, 0.25, and 2.5 cd.s/m² for scotopic phase assessments following dark adaptation, and with flash intensity

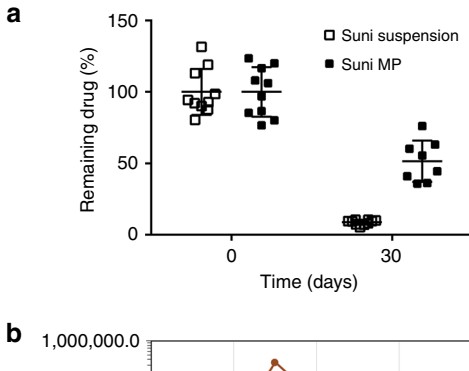

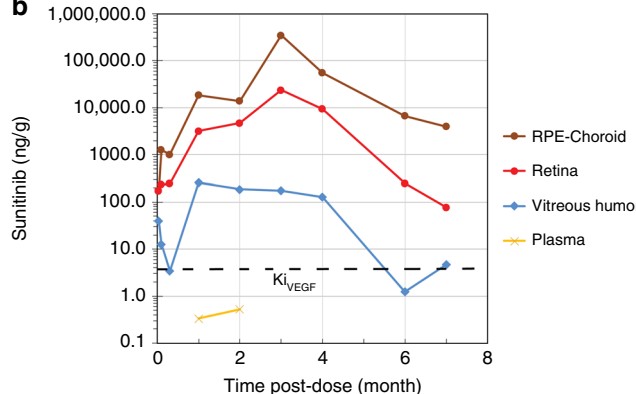

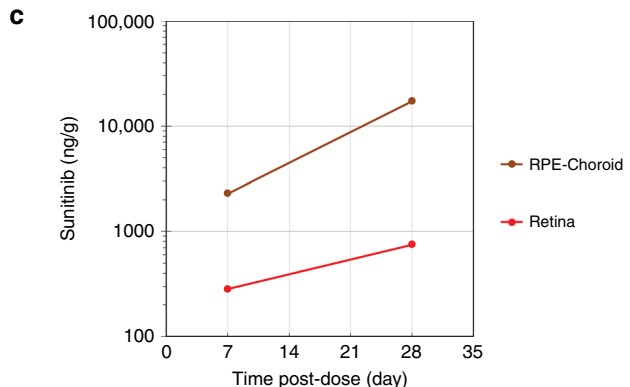

**Fig. 7 Pharmacokinetic profiles of sunitinib MPs. a** C57BL/6 mice were given a 1 µl intravitreous injection of either MPs containing 1 µg sunitinib (closed bar) or an equivalent dose of sunitinib drug suspension (open bar) in both eyes. Immediately after injection or at 30 days after injection, 5 mice from each condition were euthanized, and the amount of sunitinib remaining in the eyes was evaluated by a validated extraction and the LC–MS method and presented as a percentage of the initial dose. **b** A 50 µl suspension of sunitinib MPs containing 1 mg sunitinib was administered to both eyes of pigmented New Zealand rabbits. After 500 µL of blood was collected, animals were euthanized and the eyes were enucleated immediately after sacrifice. Central region RPE-choroid, retina, and vitreous humor in one eye of each animal were collected for pharmacokinetic ocular tissue analysis at 24 h, Days 3 and 10, and Months 1–4, 6, and 7. N = 4 biologically independent samples for time points through 4 month, n = 2 for 6 months, and n = 1 for 7 months. **c** A 200 µl suspension of sunitinib MPs containing 2 mg sunitinib was administered to both eyes of pigmented New Zealand rabbits. Central region RPE-choroid and retina in one eye of each animal were collected Days 7 and 28 (n = 3 biologically independent samples for each tissue at each time point). Sunitinib levels in plasma and ocular tissues were evaluated by LC–MS. Source data are provided as a Source Data file.

of 2.5 cd.s/m$^2$ for photopic phase. Statistically significant group × time × eye interactions were observed for ERG a- and b-wave amplitude values at scotopic 0.25 cd.s/m$^2$, b-wave amplitude values at scotopic 2.5 cd.s/m$^2$, and b-wave amplitude values at photopic 2.5 cd.s/m$^2$ only in male animals and in some occasions. These differences were interpreted to be due to inter-time point recording variability associated with sedation depth because similar reductions were observed between right and left eyes, b/a wave ratios were appropriately maintained and did not show statistically significant group × time × eye interactions, and analogous differences were not observed in females. Furthermore, there were no obvious trends across time or group which suggested dose-dependent sunitinib-MP effects on retinal function. In conclusion, repeated intravitreous injection of sunitinib over 40 weeks has no negative effects on retinal function.

There were no sunitinib MP-related gross or microscopic ocular findings other than those considered incidental and commonly observed in this strain and age of minipigs, and/or were of similar incidence in control and treated eye and, therefore, were considered unrelated to administration of sunitinib MPs.

## Discussion

Sustained suppression of VEGF signaling in the retina and choroid has the potential to improve outcomes and reduce treatment and visit burdens for patients with NVAMD. In this study, we have incorporated sunitinib, a small molecule tyrosine kinase inhibitor that blocks signaling through VEGF receptors, into biodegradable polymer MPs that after intravitreous injection provide dose-dependent, long duration suppression of type 2 choroidal NV. A single dose of MPs containing 10 µg sunitinib had persistent peak activity 6 months after injection, far longer than an intravitreous injection of 40 µg of aflibercept. While intravitreous injection is the projected route of administration, subconjunctival injection of MPs, a less invasive approach, may be an option because it provides local suppression of type 2 choroidal NV in the injected eye without an identifiable systemic effect in the fellow eye. In rho/VEGF mice, intravitreous injection of MPs containing 10 µg sunitinib provides sustained inhibition of subretinal NV for at least 4 weeks, the longest time point tested, while an intravitreous injection of 40 µg of aflibercept inhibited subretinal NV for 1 week. Much of the clinical benefit of anti-VEGF agents comes from rapid reduction of leakage from established subretinal NV and sunitinib MPs provided dramatic reduction in leakage within 1 week of injection similar to that seen with intravitreous injection of 40 µg of aflibercept. The minipig eye is closer in size to a human eye and may be more predictive for the duration of action. An intravitreous injection of sunitinib MPs resulted in a consolidated depot in the inferior vitreous base well outside the visual axis. The size of the depot gradually decreased over time and it was barely visible by Month 5 when a second injection of MPs was given. There was no identifiable ocular toxicity at Month 6, 1 month after the second injection. Six months after a 50 µl intravitreous injection of MPs containing 1 mg of sunitinib in rabbits, the drug levels were >1000-fold higher than the Ki for VEGFR2 and VEGFR1 in RPE/choroid and >50-fold higher than the Ki in retina.

In order to achieve these impressive results, a number of challenges were overcome. The three key challenges for ocular administration of MPs were inflammation from the PLGA MPs, duration of action (6 months or longer), and dispersion of MPs after injection. The pro-inflammatory effects of PLGA MPs were eliminated by the use of a hydrophilic coating (PEG) that forms on the surface of MPs when PLGA-PEG is used in the

formulation. The duration of action over 6 months was accomplished by the combination of prolonged release of sunitinib from the MPs and the binding of sunitinib to melanin granules in the RPE/choroid, which served as a secondary depot of sunitinib. Finally, surface treatment facilitated self-aggregation of MPs and the formation of a solid depot upon injection into the vitreous, thus mitigating the risk of interfering with vision by dispersed MPs.

About 20% of patients with NVAMD treated with anti-VEGF agents develop patches of macular atrophy due to photoreceptor cell death over the course of 2 years[2]. A major risk factor for the development of atrophy is the presence of type 3 choroidal NV. In *rho/VEGF* mice with type 3 choroidal NV, photoreceptor cell death resulting in retinal atrophy is due to oxidative damage and while aflibercept does not increase photoreceptor death and loss of function, it does not significantly reduce photoreceptors death or maintain rhodopsin kinase, a critical component of the visual transduction cascade that is a marker for photoreceptor survival and function[40]. In contrast, sunitinib MP promoted photoreceptor survival as demonstrated by significantly greater outer nuclear thickness and higher rhodopsin kinase level in *rho/VEGF* mice treated with sunitinib MP. This suggests that sunitinib MPs may reduce atrophy in patients with type 3 choroidal NV. This could provide an important advantage over currently available anti-VEGF agents. While the mechanism by which sunitinib promotes photoreceptor survival in *rho/VEGF* mice is uncertain, one possibility is inhibition of the dual leucine zipper kinase, an activity of sunitinib that has been demonstrated to promote survival of injured ganglion cells[46].

In addition to the major benefits in NVAMD, anti-VEGF agents provide benefits in the ischemic retinopathies, diabetic retinopathy and retinal vein occlusions. Anti-VEGF agents reduce excessive vascular leakage, reduce macular thickening, and improve vision in patients with diabetic macular edema or macular edema due to retinal vein occlusion[19–24,47–49]. The main impediment to obtaining these outstanding outcomes in clinical practice is the need for repeated intravitreous injections, a problem that could be overcome by the long duration of action of sunitinib MPs. Anti-VEGF agents have also been demonstrated to reduce retinal nonperfusion in retinal vein occlusion and diabetic retinopathy[25,26], which drives disease progression in both[28,42,50]. As a result, injections of antiVEGF agents cause regression of diabetic retinopathy[27] and the retinopathy associated with retinal vein occlusions[22,23,48,49]. Sunitinib MPs significantly reduced VEGF-induced leukocytic plugging and retinal nonperfusion. Since activation of VEGFR1 on leukocytes promotes leukocyte recruitment which contributes to leukocyte plugging[41], sunitinib MP, which block both VEGFR2 and VEGFR1 and have a long duration of action, may be ideal for treatment of diabetic retinopathy in the absence of macular edema.

In summary, we have demonstrated that a single intravitreous injection of sunitinib MPs provides prolonged suppression of type 2 and type 3 choroidal NV in mice and therapeutic levels in RPE/choroid and retina of rabbits for at least 6 months. The prolonged suppression of VEGF signaling provided by sunitinib MPs has a high chance of improving the treatment of patients with NVAMD, diabetic retinopathy, and retinal vein occlusion.

## Methods

**Preparation of drug-loaded microparticles.** Polymer MPs loaded with sunitinib were prepared using a single emulsion solvent evaporation method[33]. Briefly, 565.6 mg of PLGA 7525 4A and PLGA-PEG5k (Evonik Corporation, Piscataway, NJ) were dissolved in 4 ml dichloromethane (DCM) and mixed with a solution of 90 mg sunitinib malate (TAPI, Parsippany, NJ) in 2 ml DMSO. The mixture was homogenized in 200 ml of an aqueous solution of 1% polyvinyl alcohol (25 kDa, Polysciences, Warrington, PA) in 1× PBS using a laboratory mixer (L5M-A, Silverson Machines, East Longmeadow, MA) for 1 min. For MPs tested in mouse

models, a 1.5″ workhead was used and for those tested in rabbits, a 0.75″ workhead was used. The MPs were hardened by stirring at room temperature for 2 h to allow DCM to evaporate and then collected by sedimentation, washed thrice with cell culture grade water (HyClone, Fisher, Pittsburg, PA) and lyophilized. The dry powder of sunitinib MPs was suspended in a cold solution containing 70% ethanol and 75 mM NaOH and stirred for 3 min, collected, washed and lyophilized again. For animal studies, the lyophilized MPs were suspended in a 0.1% sodium hyaluronate solution at desired concentration prior to injection. A pulled glass micropipette or a 27-G needle was used for injections in mice and rabbits, respectively.

**Characterization of microparticles.** A Coulter MultiSizer 4 (Beckman Coulter, Inc., Miami, FL) was used to determine the size of MPs. To determine the drug loading, MPs were dissolved in DMSO and the total drug content was calculated by measuring the ultraviolet (UV) absorbance at 440 nm and comparing to a calibration curve.

**Drug release from microparticles.** Sunitinib MPs were suspended in 4 ml PBS (pH 7.4) containing 1% polysorbate 20 at 2.5 mg/ml and incubated at 37 °C on a rotating platform (150 rpm). At selected time points, 3 ml of the release medium was collected and replaced with 3 ml fresh release medium. Sunitinib content in release medium was measured by UV absorbance at 440 nm.

**Mouse model of laser-induced choroidal NV.** Wild-type female 4-week-old C57BL/6N mice were used for the experiments (Charles River, Wilmington, MA). Mice were treated in accordance with the Association for Research in Vision and Ophthalmology Guidelines on the care and use of animals in research. All protocols were approved by the Animal Care and Use Committee of the Johns Hopkins University School of Medicine. Laser photocoagulation-induced rupture of Bruch's membrane was used to generate choroidal NV[34]. Briefly, 4–5-week-old C57BL/6 mice were anesthetized, pupils were dilated with 1% tropicamide (Alcon Labs, Inc., Fort Worth, TX), and one eye was given an intravitreal injection of MPs containing 10 or 1 μg sunitinib in one eye and an equivalent mass of empty MPs in the fellow eye, or they were given an injection of 40 μg of aflibercept in one eye and PBS in the fellow eye. At various time points after injection ranging from 1 to 24 weeks after injection, the mice had rupture of Bruch's membrane by laser photocoagulation at 9, 12, and 3 o'clock positions of the posterior pole in each eye with 532 nm diode laser photocoagulation (75 μm spot size, 0.1 s duration, 120 mW) using the slit lamp delivery system of an OcuLight GL Photocoagulator (Iridex, Mountain View, CA, USA) and a hand-held cover slide as a contact lens. One week after rupture of Bruch's membrane, mice were euthanized, anterior segments, retinas, and vitreous was removed, and eye cups were stained with fluorescein isothiocyanate (FITC)-conjugated GSA (Vector Laboratories, Burlingame, CA), and flat mounted. Flat mounts were examined by fluorescence microscopy and the area of each choroidal NV was measured by image analysis with Image-Pro Plus software (Media Cybernetics, Silver Spring, MD) by an observer masked with respect to experimental groups. The three areas obtained in each eye were averaged to give a single experimental value.

In some experiments, C57BL/6 mice were given a subconjunctival injection of MPs containing 20 or 2 μg sunitinib or an equivalent mass of empty MPs in one eye and no injection in the fellow eye. At 1 week after injection, Bruch's membrane was ruptured by laser photocoagulation at three locations in each eye and after 1 week the area of choroidal NV at Bruch's membrane sites was measured.

**Transgenic mice with VEGF expression in photoreceptors.** *Rho/VEGF* transgenic mice, in which the *rhodopsin* promoter drives expression of VEGF in photoreceptors have increased levels of VEGF starting at P7 and develop multiple areas of subretinal NV by P21[37,38]. At P14, *rho/VEGF* mice of either sex, were given an intraocular injection of MPs containing 10 μg sunitinib in one eye and an equivalent mass of empty MPs in the other eye or 40 μg of aflibercept in one eye and PBS in the other eye. At P21, P28, P35, or P42, mice were euthanized and retinas were stained with FITC-labeled GSA lectin and flat mounted with photoreceptors facing up. Retinal flat mounts were examined by fluorescence microscopy and the area of subretinal NV was measured by image analysis by a masked observer.

**Fluorescein angiography.** Rho/VEGF mice were anesthetized, pupils were dilated with 1% tropicamide, and fundus photographs were obtained with a Micron III Retinal Imaging Microscope (Phoenix Research Laboratories Inc., Pleasanton, CA) before and at several time points after intraperitoneal injection of 50 μl of 25% fluorescein (AK-Fluor; Lake Forest, IL).

**Measurement of albumin in vitreous samples.** Vitreous samples were obtained by inserting a micropipette into the vitreous cavity and gently aspirating the vitreous[39]. Using the manufacturer's instructions, a mouse albumin ELISA kit (ab108791; Abcam, Cambridge, MA) was used to measure albumin levels in 1 μl of diluted vitreous humor and albumin samples for standard curve generation. The

plate wa read at 450 and 570 nm with SpectraMax Plus (Molecular Devices, San Jose, CA).

**Measurement of retinal vascular leakage with Evans Blue dye**. C57BL/6N mice, 5–7 weeks old of either sex, were given an intravitreous injection of MP containing 10 µg sunitinib in one eye and an equivalent mass of empty MPs in the fellow eye followed 1 week later by an intravitreous injection of 100 ng of VEGF in each eye. After 12 h, mice were given an intraperitoneal injection of 300 µl of Evans Blue solution (20 mg/ml; Sigma) and after 1 h they were anesthetized, the chest was opened to expose the heart, the right atrium was cut for drainage, a 27-G cannula was inserted into left ventricle, and the vasculature was flushed with PBS for 3 min. Retinas were dissected, weighed, and incubated in 150 µl formamide at 70 °C for 18 h. Retina-formamide extracts were centrifuged at $16,000 \times g$, 60 µl of supernatant was added to a 96-well plate and optical density was measured at 620 nm on a plate reader. A standard curve was used to determine the amount of Evans Blue per mg retina.

**Photoreceptor survival in mice with type 3 choroidal NV**. At P14 rho/VEGF mice were given an intravitreous injection of MPs containing 10 µg sunitinib in one eye and an equivalent mass of empty MP in the other eye. At P42, some mice were euthanized and eyes were marked at the 12:00 pole and frozen in embedding compound. ONL thickness was measured at six locations in the retina[51]. Briefly, 10 µm vertical frozen sections were cut to obtain sections from 12:00 to 6:00 through the optic nerve which were fixed in 4% paraformaldehyde, stained with hematoxylin, and examined by light microscopy. With the observer masked with respect to treatment group, ONL thickness was measured at six locations, 25% (S1), 50% (S2), and 75% (S3) of the distance between the superior pole and the optic nerve and 25% (I1), 50% (I2), and 75% (I3) of the distance between the inferior pole the optic nerve. At P49 five mice were euthanized and retinas were dissected from each eye and homogenized in 150 µl lysis buffer (Abcam, Cambridge, MA) followed by high speed centrifuge ($16,000 \times g$, 15 min). Supernatants were collected in pre-chilled eppendorf tubes. Total protein concentrations were determined using the BCA protein assay kit (BioRad, Hercules, CA) and samples containing 2 µg protein in loading buffer were resolved by 4–2% Nu-polyacrylamide gel electrophoresis (Invitrogen, Carlsbad, CA) and transferred to polyvinylidene difluoride membrane (Invitrogen) and blocked with 5% blocker milk. Membranes were probed with anti-GRK1 antibody (1:1000; ThermoFischer Scientific, Catalog no. PA5-13725, Lot no. RG22174514, Carlsbad, CA) or actin (1:2000; Catalog no. A2066, Lot no. 019M4777V, Sigma Life Science, St. Louis, MO) overnight at 4 °C. Goat anti-rabbit antibody (1:10,000 Catalog no., GENA934, Lot no. 16938431, Millipore Sigma, St. Louis, MO) was added followed by incubation in SuperSignal West Dura Extended Duration Substrate (Thermofisher), and Western blots were imaged and analyzed with a Bio-mad imaging system.

**Measurement of retinal leukostasis**. Leukostasis was measured by staining leukocytes with rhodamine- or FITC-labeled conA[52]. Briefly, mice were anesthetized, the chest was opened to expose the heart, the right atrium was cut for drainage, a 27-G cannula was inserted into left ventricle, and the vasculature was flushed with PBS for 3 min to removal all nonadherent cells. Mice were then perfused over a span of 2–3 min with rhodamine- or FITC-labeled conA (20 µg/ml in PBS, 5 mg/kg; RL-1002 and FL-1001; Vector Labs Thermofisher). The vasculature was flushed for 4 min with PBS to remove residual unbound conn A. Eyes were removed and fixed in 2% PFA for 2 h at room temperature and retinas were dissected and flat mounted. Retinas were examined by fluorescence microscopy and the number of intravascular leukocytes throughout the entire retina were counted by an investigator masked with respect to treatment group. To identify adherent leukocytes, mice were perfused with FITC-labeled conA and retinas were dissected, fixed, and permeablized with 1% Triton X-100 (Sigma) in PBS for 24 h. Nonspecific binding was blocked by 1% bovine serum albumin (Sigma) in 5% horse serum (Invitrogen). After 3 washes with PBS, retinas were incubated with rat-anti-mouse F4/80 (1:200; 123102 BioLegend) overnight at 4 °C. After washing, retinas were incubated in Alexa Fluor 594-conjugated goat anti-rat secondary antibody (1:500; A-11007 Invitrogen) for 1 h at room temperature. Retinas were washed, flat mounted and examined by fluorescence microscopy.

Leukostasis was also assessed by immunohistochemical staining to identify CD45-positive cells in retinal vessels. C57BL/6 mice of either sex at 5–7 weeks were given an intravitreous injection of MP containing 10 µg sunitinib in one eye and an equivalent mass of empty MPs in the fellow eye followed 1 week later by an intravitreous injection of 100 ng of VEGF in each eye. After 12 h, mice were euthanized, eyes were removed and fixed in 2% PFA for 2 h at room temperature, and retinas were dissected. Nonspecific binding was blocked by 1% bovine serum albumin (Sigma). After 3 washes with PBS, retinas were incubated with rat PE-conjugated anti-mouse CD45 (1:200; 12045182 Invitrogen) overnight at 4 °C. After washing, retinas were flat mounted and examined by fluorescence microscopy. The number of intravascular CD45-positive cells in retinal vessels was measured by an investigator masked with respect to treatment group.

**Pharmacokinetic studies in mice and rabbits**. Twenty male or female 5–7-week-old C57BL/6N mice were anesthetized and the pupils were dilated as previously described. Each mouse was given bilateral intravitreous injections of either MPs containing 1 µg sunitinib or an equivalent dose of sunitinib drug suspension. Immediately after injection or at 30 days after injection, 5 mice from each condition were euthanized and the whole globes were collected and transferred to a bioanalytical facility for measurement of drug levels by liquid chromatography–mass spectrometry (LC–MS). The amount of drug remaining in the eye at 30 days was compared to and presented as a percentage of the initial dose injected in the eye.

Male pigmented New Zealand rabbits (Oryctolagus cuniculus) were treated ethically in accordance with the Association for Research in Vision and Ophthalmology Guidelines on the care and use of animals in research.and protocols were approved by the Powered Research animal care and use committee. A 50 µl suspension of MPs containing 1 mg sunitinib malate was administered to both eyes through a 27 G needle. A 500 µl blood sample was collected from the rabbit's ear vein into $K_2$ EDTA tubes, the plasma separated by centrifugation and frozen at ≤70 °C. Eyes were enucleated immediately after sacrifice. Only one eye was collected for pharmacokinetic ocular tissue analysis. The following ocular tissues were collected at 24 h, Days 3 and 10, and Months 1–4, 6, and 7: central region RPE-choroid, retina, and vitreous humor. Tissue was harvested into preweighed tubes. Ocular tissue and plasma were stored frozen (at or below −70 °C) in Eppendorf tubes after collection and transferred to the bioanalytical facility at designated intervals for analysis.

For the subconjunctival injection study, a 200 µl suspension of MPs containing 2 mg sunitinib was administered to both eyes through a 27 G needle. Only one eye was collected for pharmacokinetic ocular tissue analysis. Central region RPE-choroid and retina were collected Days 7 and 28. Sunitinib levels in plasma and ocular tissues were evaluated by a validated extraction and the LC–MS method.

**Safety studies in minipigs**. Ocular tolerability and safety was tested in 15 male and 15 female 6-month-old Yucatan minipigs in an independent laboratory using Good Laboratory Practice (Charles River Laboratories, Mattawan, MI). The minipigs were treated ethically in accordance with the Association for Research in Vision and Ophthalmology Guidelines on the care and use of animals in research. The protocol was approved by the Charles River Laboratories animal care and use committee. There were four test groups (Table 1). A suspension of MPs (12–24 µl) containing 0.25, 0.5, or 1 mg sunitinib was injected into one eye and the fellow eye was injected with PBS. The first injection was given at Day 1 and the second injection was given at Day 149. Slit lamp examination, indirect ophthalmoscopy, and measurement of IOP were performed by a veterinary ophthalmologist 8, 43, 92, 142 days after each injection. ERGs were performed prior to the initiation of the study and in Weeks 19, 30, and 39. At the end of the 40-week observation period, the minipigs were euthanized and the eyes were collected.

**Statistical analyses**. No sample size calculations were done in this study. Sample sizes were estimated based upon previous similar experiments done in the past. We controlled for covariates by injecting the same animal in one eye with control and the fellow eye with the drug. Results are presented as the mean ± standard error of the mean. A Shapiro–Wilk normality test was conducted for each experiment, showing that the data did not follow a normal distribution. Therefore, statistical comparisons between two groups were done with a two-sided Mann–Whitney test

### Table 1 Ocular tolerability and safety in Yucatan minipigs.

| Group | Test material | Sunitinib malate (mg/eye) | Control | Injection volume (µl) | Males | Females |
|---|---|---|---|---|---|---|
| 1 | Sunitinib MP | 0.25 | PBS | 12 | 3 | 3 |
| 2 | Sunitinib MP | 0.5 | PBS | 12 | 3 | 3 |
| 3 | Sunitinib MP | 0.5 | PBS | 24 | 3 | 3 |
| 4 | Sunitinib MP | 1.0 | PBS | 24 | 3 | 3 |

A suspension (12–24 µl) of sunitinib microparticles (MPs) containing 0.25, 0.5, or 1 mg of sunitinib was injected into one eye and the fellow eye was injected with phosphate-buffered saline (PBS). Injections were repeated on Day 149. Slit lamp examination, indirect ophthalmoscopy, and measurement of IOP were performed by a veterinary ophthalmologist 8, 43, 92, 142 days after each injection. ERGs were performed prior to the initiation of the study and in Weeks 19, 30, and 39. At week 40, minipigs were euthanized, eyes were removed, and histopathologic examination was done.

and comparisons between more than two groups were done by Kruskal–Wallis test followed by Dunn's test using GraphPad Prism software (GraphPad Software, La Jolla, CA).

**Reporting summary**. Further information on research design is available in the Nature Research Reporting Summary linked to this article.

## Data availability

All data generated during this study are included in this published article (and its Supplementary Information files). The supplementary source data file contains the source data underlying Figs. 2a–d, 3a, b, 4a–e, 5b–d, 6a, b, and 7a–c.

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

## Acknowledgements

Graybug Vision; Altsheler-Durell Foundation, Research to Prevent Blindness, New York, NY.

## Author contributions

H.T. performed the experiments, collected and analyzed the data, prepared figures, edited and approved the paper. J.F. designed the microparticles, synthesized the microparticles, performed the experiments, collected and analyzed the data, edited and approved the paper. J.S. performed the experiments, collected the data, analyzed the data, edited and approved the paper. Y.Y. improved and synthesized the microparticles, performed the experiments, collected and analyzed the data, edited and approved the paper. Z.H. assisted in performance of the experiments and collected the data. J.K. improved and synthesized the microparticles, performed the experiments, collected and analyzed the data, edited and approved the paper. D.M. synthesized the microparticles, performed the experiments, collected and analyzed the data, edited and approved the paper. D.Ca. synthesized the microparticles, performed the experiments, collected and analyzed the data, edited and approved the paper. D.C. designed and supervised the experiments, edited and approved the paper. W.P. designed and supervised the experiments, analyzed the data, edited and approved the paper. B.C.G. performed the experiments, edited and approved the paper. C.C. analyzed the data, edited and approved the paper. J.Z. designed and supervised the experiments, analyzed the data, edited and approved the paper. Y.K. performed the experiments, collected and analyzed the data, edited and approved the paper. W.Y. analyzed the data, edited and approved the paper. J.L.C. provided resources, designed the experiments, edited and approved the paper. M.Y. designed improvements in microparticles, designed and supervised the experiments, analyzed the data, wrote portions of the paper, edited and approved the paper. J.H. designed the microparticles, designed and supervised the experiments, analyzed the data, edited and approved the paper. P.A.C. designed and supervised the experiments, analyzed the data, wrote first draft of the paper, edited and approved the paper.

## Competing interests

J.H., J.F. and P.A.C. are founders of Graybug Vision which has a commercial interest in sunitinib microparticles, and they have equity in Graybug and receive remuneration for consultation. Johns Hopkins University has patents on which J.H. and J.F. are authors that are licensed to Graybug Vision. All of these relationships are being managed by the Johns Hopkins University conflict of interest committee. Y.Y., D.M., D.Ca., W.Y. and M.Y. are employees of Graybug Vision. J.K., W.P., J.Z.Z. and J.L.C. are former employees of Graybug Vision. The remaining authors declare no competing interests.
