## [Peer Review File · Nature Communications]

Reviewers' comments:

Reviewer #1 (Remarks to the Author):

The authors should be congratulated for assessing their TKI and delivery system in a variety of animal models that are relevant for the back of the eye diseases. Tyrosine kinase inhibitors have been previously shown to be effective in similar small as well as large animal models of relevance for the back of the eye diseases. Also, some of them have been assessed in clinical trials, but failed to show significant efficacy, relative to the standard of care. Although an anti-VEGF protein drug (targeting human protein) was used for comparison purposes in this study, their effects in some of the animal models are limited. Drug binding to tissues might be one reason for sustained effects, as previously reported in the literature for TKIs. While the collective set of experiments and the delivery system, albeit based on very routinely used polymers, are interesting, there is limited innovation in the present study. Further the findings are not broadly relevant to the readership. The paper is more suited for an ophthalmology journal.

Reviewer #2 (Remarks to the Author):

This article provides compelling data for a new delivery system for an established tyrosine kinase inhibitor, sunitinib, for blinding eye diseases and tests the delivery device in models of wet AMD. Specifically the investigators test the sunitinib microparticles in models of choroidal neovascularization, vessel permeability and leukostasis as well as rod loss. Overall, the data provide strong evidence for a significantly extended time course of inhibition of VEGF signaling that may be achieved by the delivery of sunitinib microparticle, potentially reducing frequency of intravitreal or subconjunctival injections. The study is highly significant but does not provide any additional mechanistic data the disease or how the drug acts.

The data are compelling for methods measuring neovascularization. However, there are concerns regarding leukostasis and vascular permeability measures.

For vascular permeability, figure 4 C and unlabeled ELISA; these measures of permeability are not very convincing. The fluorescein images of SuniMP vs EmptyMP after 7 days do not demonstrate a clear difference in leak of fluorescein.

The ELISA for albumin needs control animals without induction of VEGF expression.

The Elisa assay is useful but dependent on a host of variables including albumin turnover rate and clearance. Please provide a measure of Evan 's blue permeability with a defined circulation time that would better provide a direct measure of albumin permeation of the vessels.

For leukostasis: Figure 6 use of concanavalin A is not sufficient for a journal of this quality. Immunostaining of specific inflammatory cell markers such as CD45, cd11 etc with high resolution confocal microscopy, or even better, flow analysis for specific inflammatory cells is needed to make a statement about leukostasis or infiltration. The current method makes no distinction of types of inflammatory cells or recruited angioblasts. Further, the images provided are highly limited.

Again, no control animals without VEGF are provided as a baseline.

Also, Ocular tolerability: no data was given, just a statement that the drug was well tolerated. What about an OCT scan for structure? ERG for light response? Without data this section needs to be removed.

Additional specific comments are provided below.

Abstract, "leukocytic plugging" do you mean leukostasis? Please revise.

Introduction:

"and slow death of photoreceptors, RPE, and choriocapillaris in the macula" should read "and choroidal capillaries beneath the macula resulting in gradual loss of central vision.

"The most common cause of vision loss in each is macular edema due to leakage from retinal blood vessels." Please provide reference.

"These two effects of VEGF suppression are related because slowly progressive closure of retinal vessels is the underlying cause of progression of diabetic retinopathy (25)." This statement is debatable and ignores the effect of VEGF on vascular permeability as a cause for DME and diabetic retinopathy. Blocking VEGF alleviates vessel closure but was that causative in reducing DME or was reduced vessel permeability? Please revise. In general, scientific writing refrains from the declarative "causes", which is used liberally here, unless definitive causal, not correlative, evidence is provided.

Figure legend 1 does not give enough information as to how the release experiment was completed in 1a. Is this an intravitreal injection? Is the animal sacrificed at each time point? How many animals used? How was drug release measured? Was remaining drug in the pellet measured?

Results "Intravitreal injections of anti-VEGF agents are better tolerated than clinicians imagined possible prior to 2006, but they are associated with discomfort and anxiety which reduces compliance and compromises visual outcomes. If a less invasive mode of administration was possible, it would be enthusiastically embraced by patients and improve compliance." None of this conjecture is appropriate for Results section.

Figure 2 and 4, since each condition is compared only to its control at the same time, the graph should be a bar graph with comparison at each time point and not a graph connecting the time points.

Kruskal Wallis rank measure does not seem appropriate for fig3. Why not use ANOVA?

Figure 4: The ELISA graph should be labeled 4D.

Results: Remove the sentence: "This indicates that the photoreceptor cell death that occurs in eyes with type 3 choroidal NV occurs from oxidative damage and has nothing to do with VEGF suppression." As written, the sentence does not make sense in the context of the paragraph. Further, the negative result with anti-VEGF does not allow the conclusion reached.

Results: The phrase "leukocyte plugging" is again used. Please revise.

Also, regions of hypoxia due to leukostasis have not been definitively demonstrated as the main driver of diabetic retinopathy and a large number of competing hypothesis remain viable. Indeed, it remains unclear if all patients with DR have the same disease etiology. Please revise.

Figure 7: Given the low plasma concentration, how do you suppose the sunitinib gets to the RPE-choroid after intravitreal injection? Is there data to support a route to the RPE/choroid?

Discussion: The authors state that the microparticles did not yield any inflammation but no data is given to support this statement.

Reviewer #3 (Remarks to the Author):

The authors modified the anti-cancer drug Sunitinib coated with PLGA-PEG to extend anti-angiogenesis function in several different ocular neovascular disease animal models. Authors demonstrated that this may suppress choroidal and retinal neovascularization, retinal leakage, and adherent intravascular leukocytes and improves retinal perfusion in mouse model. T

Major comments:

1. Please explain more detail why choose sunitinib? Authors only indicated that sunitinib may bind VEGF receptor 1 and 2. However, it also inhibits PDGFR- β receptor. Please compare pros and cons to aflibercept.

2. It is known that long term angiogenesis inhibition in the retina may raise risk for neuronal atrophy. In this paper, authors don't show any safety data. Please show data on retinal function and thickness after administration of sunitinib

3. Please explain why microparticles are necessary for sunitinib. This question arise from the reasons below.

- PLGA microparticle is biodegradable. In fact, the author showed it in Figure 1B. 50 μ L of microparticle almost disappear on day 92 in a rabbit eye.

- I think up to 1 μ L is the volume of the intravitreal injection. Just in case, I assumed 1 μ L contain the microparticle at 100%. As mentioned in the Page7, I understand the real microparticle volume is much less than 1 μ L.

- By rough estimation, in rabbit eye, 50% of microparticle (25 μ L) dissolved in 100 days. If so, the microparticle dissolve at 0.25 μ L per day. 1 μ L microparticle can dissolve within 1-2 week.

- However, the author showed sunitinib microparticle suppressed laser induced CNV 24 weeks post injection in figure 2A. I don't think 1 μ L biodegradable microparticle stays there for such long time by the above estimation.

- If so, why did it work 24 weeks post injection? I think this is due to low solubility of sunitinib at more than neutral pH.

- After sunitinib microparticle injection, PLGA microparticle may disappear. Then, it may leave sunitinib pellet.

- Sunitinib pellet would maintain its functional level in the eye by slowly dissolving.

- If one injects sunitinib suspension in a hyaluronate solution, we may get the same result with this paper.

Thus, I think this paper fails to show significant role of microparticle for releasing hydrophobic (pH dependency) drug.

To overcome this problem, the easiest way is to show the microparticle in the mouse eye 20-24 weeks post injection. However, if 1 μ L microparticle exist in the eye after 24 weeks, we can say this microparticle is not biodegradable.

Minor comments:

3. Please indicate how much volume injected for mouse vitreous and conjunctiva.

Dear Reviewers,

Thank you for evaluating our manuscript “Sustained Treatment of Retinal Vascular Diseases with Self-Aggregating Sunitinib Microparticles.” There were many useful suggestions that have helped us to extensively revise and improve the manuscript. The following is a point-by-point response to each of the comments and an explanation as to how the manuscript has been extensively revised.

Reviewers' comments:

Reviewer #1 (Remarks to the Author):

The authors should be congratulated for assessing their TKI and delivery system in a variety of animal models that are relevant for the back of the eye diseases. Tyrosine kinase inhibitors have been previously shown to be effective in similar small as well as large animal models of relevance for the back of the eye diseases. Also, some of them have been assessed in clinical trials, but failed to show significant efficacy, relative to the standard of care. Although an anti-VEGF protein drug (targeting human protein) was used for comparison purposes in this study, their effects in some of the animal models are limited. Drug binding to tissues might be one reason for sustained effects, as previously reported in the literature for TKIs. While the collective set of experiments and the delivery system, albeit based on very routinely used polymers, are interesting, there is limited innovation in the present study. Further the findings are not broadly relevant to the readership. The paper is more suited for an ophthalmology journal.

There have been previous attempts to provide sustained delivery of medications to the retina by loading them in polymers, making microparticles or nanoparticles, and injecting them into the eye. All of these attempts have failed because of severe inflammation and/or dispersion of the particles throughout the eye degrading vision or causing glaucoma. Since the current manuscript describes a successful approach to the use of microparticles to provide long term benefit in a highly prevalent disease without the problems that have previously caused failure, it is novel. The Introduction explains that in clinical practice, patients with neovascular age-related macular degeneration (NVAMD) are not receiving the frequency of antiVEGF injections that are needed and visual outcomes are much poorer than those obtained in clinical trials; therefore, longer-acting treatments are an unmet medical need. This manuscript describes a successful strategy to overcome this important unmet medical need. Major companies that address a large variety of diseases, in addition to NVAMD, have abandoned programs aimed at doing what is described in this manuscript because they viewed the problems as unable to be solved and therefore it will be interesting to a broad audience to see a solution. In addition, sustained delivery of drugs is an unmet need in diseases of other organ systems and the procedures described in this manuscript may be helpful to clinicians and scientists dealing with those diseases. Therefore we respectfully request that you consider this perspective and acknowledge that success in an endeavor for which there has always been failure in the past is worthy of communicating to the scientific and medical community.

Reviewer #2 (Remarks to the Author):

This article provides compelling data for a new delivery system for an established tyrosine kinase inhibitor, sunitinib, for blinding eye diseases and tests the delivery device in models of wet AMD. Secifically the investigators test the sunitinib microparticles in models of choroidal neovascularization, vessel permeability and leukostasis as well as rod loss. Overall, the data provide strong evidence for a significantly extended time course of inhibition of VEGF signaling that may be achieved by the delivery of sunitinib microparticle, potentially reducing frequency of intravitreal or subconjunctival injections. The study is highly significant but does not provide any additional mechanistic data the disease or how the drug acts.

Thank you for providing the perspective that is discussed above in the response to Reviewer 1; the study is high impact because sustained suppression of VEGF is a major unmet medical need and the manuscript describes the first success using microparticles to provide sustained delivery of a TKI that blocks VEGF receptors. We submit to you that the lack of mechanistic data regarding how sunitinib suppresses choroidal neovascularization (NV) is not a weakness of this study, because this information is already well-established in the literature. In fact, work from our laboratory first demonstrated that VEGF is a critical stimulator for choroidal NV and that neutralization of VEGF or blocking phosphorylation of VEGF

receptors by systemic administration of an antiVEGF TKI strongly suppresses choroidal NV (1-4). Clinical trials have confirmed the importance of VEGF and led to approval of intravitreal injection of VEGF-neutralizing proteins for the treatment of choroidal NV in NVAMD (5). So I hope you agree that the role of VEGF signaling in choroidal NV is very well established and the mechanism by which sunitinib microparticles exert their effect is clear and requires no additional investigation.

The data are compelling for methods measuring neovascularization. However, there are concerns regarding leukostasis and vascular permeability measures.

For vascular permeability, figure 4 C and unlabeled ELISA; these measures of permeability are not very convincing. The fluorescein images of SuniMP vs EmptyMP after 7 days do not demonstrate a clear difference in leak of fluorescein.

In the original and current Figure 4C, the upper left panel shows the pretreatment fluorescein angiogram in the eye of a P28 *rho/VEGF* mouse with severe subretinal NV. There are large blotches of green fluorescence because fluorescein has leaked into the retinal tissue. The bottom left panel shows the same eye 7 days after injection of sunitinib MPs and many of the large collections of fluorescein are markedly reduced although there are still a few small spots. This clearly demonstrates that the sunitinib MPs reduced fluorescein leakage. In the original Figure 4C, the upper right panel showed the pretreatment fluorescein angiogram in another P28 *rho/VEGF* mouse with severe subretinal NV and the bottom right panel showed the same eye 7 days after injection of empty MPs. There was no reduction in extravascular fluorescein but the upper right photo was a bit out of focus which made the comparison more difficult. We have therefore replaced this control eye pair with another in which both images are in good focus and it is clear that there is no difference in the amount of fluorescein that has collected in the retina at both time points (in fact, there appears to be slightly more leakage in the post-injection photo).

The ELISA for albumin needs control animals without induction of VEGF expression.

We have measured vitreous albumin level in untreated control mice and these data have been added to Figure 4D.

The Elisa assay is useful but dependent on a host of variables including albumin turnover rate and clearance. Please provide a measure of Evan 's blue permeability with a defined circulation time that would better provide a direct measure of albumin permeation of the vessels.

The increase in serum albumin within the vitreous or retina as a measure of vascular permeability is well-validated (6) and is used by other investigators (7), but additional validation is useful. Therefore we have done the recommended additional experiments using Evans blue dye and the results are reported in Figure 4D. They also show a significant reduction in vascular leakage in eyes injected with sunitinib MPs compared with those injected with empty MPs.

For leukostasis: Figure 6 use of concanavalin A is not sufficient for a journal of this quality. Immunostaining of specific inflammatory cell markers such as CD45, cd11 etc with high resolution confocal microscopy, or even better, flow analysis for specific inflammatory cells is needed to make a statement about leukostasis or infiltration. The current method makes no distinction of types of inflammatory cells or recruited angioblasts. Further, the images provided are highly limited. Again, no control animals without VEGF are provided as a baseline.

We have previously described the types of leukocytes involved in VEGF-induced leukostasis and the molecular mechanism by which it occurs (8), so it would not be useful to repeat those sorts of experiments in this manuscript in which we are simply showing that sunitinib MPs reduce VEGF-induced leukostasis. Perfusion with concanavalin A is well accepted and widely used for assessment of leukostasis, but it is potentially useful to demonstrate things in more than one way. Therefore, we have taken the recommendation and done additional studies using staining for CD45 and the results are provided in Figure 6B. We have also done measurements in control, untreated mice and included those data.

Also, Ocular tolerability: no data was given, just a statement that the drug was well tolerated. What about an OCT scan for structure? ERG for light response? Without data this section needs to be removed.

The sunitinib microparticles have been tested in an independent laboratory using Good Laboratory Practice and have been found to be safe and well-tolerated. There was no reduction in ERG amplitudes in sunitinib MP-injected eyes and retinal histopathology showed no retinal thinning or any other signs of retinal toxicity. We have provided this information in the Results section of the revised manuscript.

Additional specific comments are provided below.

Abstract, “leukocytic plugging” do you mean leukostasis? Please revise.

This has been revised as requested.

Introduction:

“and slow death of photoreceptors, RPE, and choriocapillaris in the macula” should read “and choroidal capillaries beneath the macula resulting in gradual loss of central vision.

Actually, choriocapillaris is the preferred and most widely used term for this capillary bed. We can change it to choroidal capillaries if there is a strong preference, but most investigators use choriocapillaris.

“The most common cause of vision loss in each is macular edema due to leakage from retinal blood vessels.” Please provide reference.

References have been added.

“These two effects of VEGF suppression are related because slowly progressive closure of retinal vessels is the underlying cause of progression of diabetic retinopathy (25).” This statement is debatable and ignores the effect of VEGF on vascular permeability as a cause for DME and diabetic retinopathy. Blocking VEGF alleviates vessel closure but was that causative in reducing DME or was reduced vessel permeability? Please revise. In general, scientific writing refrains from the declarative “causes”, which is used liberally here, unless definitive causal, not correlative, evidence is provided.

We agree that VEGF-induced excessive vascular permeability is the cause of diabetic macular edema and did not say otherwise. The sentence quoted above says that progressive capillary closure is the underlying cause of progression of diabetic retinopathy which is the conclusion of the cited article. To avoid confusion, we have revised the sentence to say “background diabetic retinopathy”. To address the valid concern that association does not prove causality, we have replaced “underlying cause” with “strongly associated with”.

Figure legend 1 does not give enough information as to how the release experiment was completed in 1a. Is this an intravitreal injection? Is the animal sacrificed at each time point? How many animals used? How was drug release measured? Was remaining drug in the pellet measured?

We have revised the legend to be clearer. The procedure is described in the Methods section and is pasted here for convenience.

“Sunitinib MPs were suspended in 4 mL phosphate buffered saline (PBS, pH7.4) containing 1% polysorbate 20 at 2.5 mg/mL and incubated at 37°C on a rotating platform (150 rpm). At selected time points, 3 mL of the release medium was collected and replaced with 3 mL fresh release medium. Sunitinib content in release medium was measured by UV absorbance at 430 nm.” At the end of the experiment there was no measureable sunitinib in the remaining pellets and hence as shown the percentage released reached 100%.

Results ”Intravitreal injections of anti-VEGF agents are better tolerated than clinicians imagined possible prior to 2006, but they are associated with discomfort and anxiety which reduces compliance and compromises visual outcomes. If a less invasive mode

of administration was possible, it would be enthusiastically embraced by patients and improve compliance.” None of this conjecture is appropriate for Results section.

Agreed- this is not appropriate for the Results section and has been removed.

Figure 2 and 4, since each condition is compared only to its control at the same time, the graph should be a bar graph with comparison at each time point and not a graph connecting the time points.

We respectfully disagree. These figures show the mean area of choroidal NV for the experimental group and the control group at several time points. The lines between the points simply assist the reader to see the change in mean area over time which is helpful. This is a standard way to illustrate data from a time course experiment, but if you feel strongly, we can change to bar graphs.

Kruskal Wallis rank measure does not seem appropriate for fig3. Why not use ANOVA?

We determined that data do not have a normal distribution and therefore we could not use ANOVA which is a parametric test, but instead had to use Kruskal Wallis which is nonparametric and is more stringent.

Figure 4: The ELISA graph should be labeled 4D.

This has been done.

Results: Remove the sentence: “This indicates that the photoreceptor cell death that occurs in eyes with type 3 choroidal NV occurs from oxidative damage and has nothing to do with VEGF suppression.” As written, the sentence does not make sense in the context of the paragraph. Further, the negative result with anti-VEGF does not allow the conclusion reached.

We have revised the sentence as requested.

Results: The phrase “leukocyte plugging” is again used. Please revise.

This has been revised.

Also, regions of hypoxia due to leukostasis have not been definitively demonstrated as the main driver of diabetic retinopathy and a large number of competing hypothesis remain viable. Indeed, it remains unclear if all patients with DR have the same disease etiology. Please revise.

We have changed “the main driver of” to “which is strongly associated with” which is consistent with the two cited articles.

Figure 7: Given the low plasma concentration, how do you suppose the sunitinib gets to the RPE-choroid after intravitreal injection? Is there data to support a route to the RPE/choroid?

The sunitinib diffuses from the vitreous into the retina and RPE/choroid. It is then taken up by choroidal vessels and enters the systemic circulation, but it is a small molecule that is rapidly cleared from the systemic circulation and thus the plasma levels are very low.

Discussion: The authors state that the microparticles did not yield any inflammation but no data is given to support this statement.

We have greatly expanded this section of the Results to provide more from the GLP toxicity study. Serial in life ocular examinations and ocular histopathology at the end of the study showed no inflammation related to sunitinib MPs.

Reviewer #3 (Remarks to the Author):

The authors modified the anti-cancer drug Sunitinib coated with PLGA-PEG to extend anti-angiogenesis

function in several different ocular neovascular disease animal models. Authors demonstrated that this may suppress choroidal and retinal neovascularization, retinal leakage, and adherent intravascular leukocytes and improves retinal perfusion in mouse model.

Major comments:

1. Please explain more detail why choose sunitinib? Authors only indicated that sunitinib may bind VEGF receptor 1 and 2. However, it also inhibits PDGFR- β receptor. Please compare pros and cons to aflibercept.

We chose sunitinib because: (1) It is an approved drug and therefore its systemic safety profile with large systemic doses is well-established. The systemic exposure after intravitreal injection of sunitinib MPs is far below those previously shown to have no adverse effects. Having all the systemic safety data in FDA files accelerated the move into clinical trials.

(2) It is an excellent inhibitor of VEGF signaling and in previous studies we have demonstrated that analogs of sunitinib when given by repeated subcutaneous injections, strongly inhibit choroidal neovascularization with no identifiable toxicity (9).

2. It is known that long term angiogenesis inhibition in the retina may raise risk for neuronal atrophy. In this paper, authors don't show any safety data. Please show data on retinal function and thickness after administration of sunitinib.

We have added much more detail from a GLP toxicity study in which ERGs and histology was done after 2 injections of sunitinib MPs 20 weeks apart. The ERGs showed no reduction in retinal function and the histology showed no thinning of the retina or retinal atrophy.

3. Please explain why microparticles are necessary for sunitinib. This question arise from the reasons below.

- PLGA microparticle is biodegradable. In fact, the author showed it in Figure 1B. 50 μ L of microparticle almost disappear on day 92 in a rabbit eye.

- I think up to 1 μ L is the volume of the intravitreal injection. Just in case, I assumed 1 μ L contain the microparticle at 100%. As mentioned in the Page7, I understand the real microparticle volume is much less than 1 μ L.

- By rough estimation, in rabbit eye, 50% of microparticle (25 μ L) dissolved in 100 days. If so, the microparticle dissolve at 0.25 μ L per day. 1 μ L microparticle can dissolve within 1-2 week.

- However, the author showed sunitinib microparticle suppressed laser induced CNV 24 weeks post injection in figure 2A. I don't think 1 μ L biodegradable microparticle stays there for such long time by the above estimation.

- If so, why did it work 24 weeks post injection? I think this is due to low solubility of sunitinib at more than neutral pH.

- After sunitinib microparticle injection, PLGA microparticle may disappear. Then, it may leave sunitinib pellet.

- Sunitinib pellet would maintain its functional level in the eye by slowly dissolving.

- If one injects sunitinib suspension in a hyaluronate solution, we may get the same result with this paper.

Thus, I think this paper fails to show significant role of microparticle for releasing hydrophobic (pH dependency) drug.

To overcome this problem, the easiest way is to show the microparticle in the mouse eye 20-24 weeks post injection. However, if 1 μ L microparticle exist in the eye after 24 weeks, we can say this microparticle is not biodegradable.

We performed additional experiments to test the hypothesis that it is not necessary to incorporate sunitinib into MPs but rather that intraocular injection of free sunitinib would provide similar long-lasting efficacy.

(1) First we compared *in vitro* release of 1 mg of sunitinib suspension versus sunitinib loaded MPs at 37°C under sink conditions. The curves for *in vitro* release are shown in new Figure 1B of the revised manuscript and demonstrate that all of the free sunitinib was released by 24 days when only 34% of the

sunitinib had been released from MPs. Therefore, any depot effect related to poor aqueous solubility of sunitinib is minor compared to the depot effect of the MPs.

(2) Second we compared *in vivo* release in mouse eyes of free sunitinib suspension versus MPs loaded with an equivalent amount of sunitinib (n=20 eyes for each). Ten eyes for each group were used to measure the amount of sunitinib that was present in eyes immediately after injection at time 0. After 30 days, the remainder of the mice were euthanized and the amount of sunitinib in each eye was measured. The level of sunitinib was measured in whole eye homogenates by LC/MS. The results are shown in new Figure 7A and demonstrate that compared with the amount of sunitinib in the eye at time 0, the amount remaining at 30 days after injection was 52% for sunitinib MPs, versus 9% for free sunitinib. Thus, in order to maintain high levels of sunitinib in the eye for long duration, it is necessary to incorporate sunitinib into MPs; it cannot be achieved by injecting free sunitinib into the eye.

(3) Third, we tested the efficacy of free sunitinib. In the studies reported in this manuscript, we injected MPs containing 10 µg or 1 µg of sunitinib, so we first tested the effect of injecting 10 µg of free sunitinib. We wanted to have adequate experimental numbers and therefore injected 30 eyes with 10 µg of sunitinib and 30 eyes with PBS planning to rupture Bruch's membrane to assess effect on choroidal NV. However, 20 of the eyes injected with sunitinib had a dense cataract as shown below while 29 of the eyes injected with PBS had a completely clear lens and one had a very mild cataract.

We tested lower doses of sunitinib and found that the highest dose that caused no cataracts was 0.5 µg. We then compared intravitreal injection of 0.5 or 0.1 µg sunitinib versus PBS and found that one week after injection, there was no significant difference in area of choroidal NV. These new data have been included in Figure 2 in the revised manuscript and they indicate that the hypothesis is incorrect. The sustained delivery of low levels of sunitinib from MPs is clearly different from simply injecting sunitinib into the eye which has no efficacy at nontoxic doses, while injection of sunitinib MPs provides prolonged efficacy at nontoxic doses.

Minor comments:

3. Please indicate how much volume injected for mouse vitreous and conjunctiva.

The volume of intravitreal injections in mice was 1 µl and the volume for the subconjunctival injections was 2 µl.

We greatly appreciate the careful reviews and valuable suggestions that have helped us to improve the manuscript. The revised manuscript will provide a much clearer picture of the value of sunitinib MPs to the readers and hopefully stimulate additional work in this important area.

Sincerely,

Peter A. Campochiaro

1. Kwak N, Okamoto N, Wood JM, and Campochiaro PA. VEGF is an important stimulator in a model of choroidal neovascularization. *Invest Ophthalmol Vis Sci.* 2000;41(10):3158-64.

2. Saishin Y, Saishin Y, Takahashi K, Lima Silva R, Hylton D, Rudge J, J. WS, and Campochiaro PA. VEGF-TRAP_{R1R2} suppresses choroidal neovascularization and VEGF-induced breakdown of the blood-retinal barrier. *J Cell Physiol*. 2003;195(2):241-8.
3. Akiyama H, Mohamedali K, Lima-Silva R, Kachi S, Shen J, Hatara C, Umeda N, Hackett SF, Aslam S, Krouse M, et al. Vascular targeting of ocular neovascularization with a VEGF121/Gelatin chimeric protein. *Mol Pharmacol*. 2005;68(15):43-50.
4. Takahashi K, Saishin Y, Saishin Y, King A, Levin R, and Campochiaro PA. The multi-targeted kinase inhibitor pazopanib causes suppression and regression of choroidal neovascularization. *Arch Ophthalmol*. 2009;127(4):94-9.
5. Campochiaro PA, Aiello LP, and Rosenfeld PJ. Anti-vascular endothelial growth factor agents in the treatment of retinal disease. From bench to bedside. *Ophthalmology*. 2016;123 (10S):S78-S88.
6. Fortmann SD, Lorenc VE, Shen J, Hackett SF, and Campochiaro PA. Mousetap, a novel technique to collect uncontaminated vitreous or aqueous and expand usefulness of mouse models. *Sci Rep*. 2018;8(1):6371.
7. Thounaojam MC, Powell FL, Patel SR, Gutsaeva dR, Tawfik A, Smith SB, Nussbaum J, L. BN, Martin PM, Schally AV, et al. Protective effects of agonists of growth hormone-releasing hormone (GHRH) in early experimental diabetic retinopathy. *Proc Natl Acad Sci USA*. 2017;114(50):13248-53.
8. Liu Y, Shen J, Fortmann SD, Wang J, Vestweber D, and Campochiaro PA. Reversible retinal vessel closure from VEGF-induced leukocyte plugging. *JCI Insight*. 2017;2(18):pii: 95530. doi: 10.1172/jci.insight.
9. Miki A, Ueno S, Wesinger DM, Berlinicke C, Shaw GC, Usui S, Wang Y, Zack D, and Campochiaro PA. Prolonged blockade of VEGF receptors does not damage retinal photoreceptors or ganglion cells. *J Cell Physiol* 2010;224(1):262-72.

Reviewers' comments:

Reviewer #2 (Remarks to the Author):

The investigators explore the use of PLGA-PEG microparticles (MP) to delivery sunitinib, a tyrosine kinase inhibitor, in models of VEGF driven retinal vascular pathology. The investigators provide impressive data demonstrating the extended release and effectiveness of the sunitinib from MP in assays of both vascular angiogenesis and vascular permeability. The investigators have added significant new data to address all previous questions from this reviewer and have clarified all questions. These importantly include the Yucatan minipig studies of no-observed-adverse-effect-level. The investigators do mention a statistical difference in group*time*eye interactions and while an explanation is provided, the actual difference is never shown. Please either state the difference in Results or add a supplemental figure.

Minor comment: The phrase "choriocapillaris in the macula" just needs to state "choriocapillaris beneath the macula"

Reviewer #3 (Remarks to the Author):

The authors have made substantial improvements. But some significant questions remain.

A. 30 days post injection, 9% free sunitinib are remained in the eye (response page 6). From this, we can extrapolate that almost 8 % of sunitinib exits the eye ($0.09^{(1/30)}$). Anyway, we can estimate ~280ng sunitinib (1 week post 500ng sunitinib injection, $500 \times 0.92^{(7)}$) and 160ng sunitinib (2 weeks post 500ng sunitinib injection, $500 \times 0.92^{(14)}$) remains in the eye in an available form.

From Figure 1A-B and Figure 2A-B, we can assume 1% sunitinib are released from MP to the eye. For simplification, I assume that 20ng sunitinib/day release from 2ug sunitinib MP. If so, 140ng (1week post MP injection) and 280ng (2weeks) sunitinib would be available in the eye. However, we have to incorporate 8% loss of free sunitinib per day. If so, 100ng (1week) and 160ng (2weeks) sunitinib would be available in the eye which was injected with 2ug sunitinib MP. These calculations indicated that 500ng free sunitinib injection and 2ug sunitinib MP injection would provide very close amount of free sunitinib 1-2weeks post injection. However, 500ng sunitinib injection did not suppress laser CNV (Figure 2C), and 2ug sunitinib MP injection suppresses laser CNV (Figure 3B and others). How can you explain this contradiction? Are the microparticles anti-angiogenic in some way?

2. Also, the author should show the rate of cataract by Sunitinib MP injection to prove that MP prevent sunitinib induced cataract (response page 6).

October 14, 2019

Revision of NCOMMS-18-5508289B-Z

Dear Reviewers,

We thank you for evaluating our revised manuscript. The following is a point-by-point response to each of the comments.

Reviewer #2 (Remarks to the Author):

The investigators explore the use of PLGA-PEG microparticles (MP) to delivery sunitinib, a tyrosine kinase inhibitor, in models of VEGF driven retinal vascular pathology. The investigators provide impressive data demonstrating the extended release and effectiveness of the sunitinib from MP in assays of both vascular angiogenesis and vascular permeability. The investigators have added significant new data to address all previous questions from this reviewer and have clarified all questions. These importantly include the Yucatan minipig studies of no-observed-adverse-effect-level. The investigators do mention a statistical difference in group*time*eye interactions and while an explanation is provided, the actual difference is never shown. Please either state the difference in Results or add a supplemental figure.

As suggested by the Reviewer, we have edited the text to further explain the ERG findings. Please note the statement about electroretinography (ERG) in minipigs is taken directly from the final study report written by the Study Director of the CRO (Charles River Laboratories, Mattawan, Michigan). As quoted from the study report, "there were no obvious trends across time or group which suggested dose-dependent test-article effects on retinal function".

Minor comment: The phrase "choriocapillaris in the macula" just needs to state "choriocapillaris beneath the macula"

This has been changed as requested.

Reviewer #3 (Remarks to the Author):

The authors have made substantial improvements. But some significant questions remain.

A. 30 days post injection, 9% free sunitinib are remained in the eye (response page 6). From this, we can extrapolate that almost 8 % of sunitinib exits the eye ($0.09^{(1/30)}$). Anyway, we can estimate ~280ng sunitinib (1 week post 500ng sunitinib injection, 500×0.92^7) and 160ng sunitinib (2 weeks post 500ng sunitinib injection, 500×0.92^{14}) remains in the eye in an available form.

From Figure1A-B and Figure 2A-B, we can assume 1% sunitinib are released from MP to the eye. For simplification, I assume that 20ng sunitinib/day release from 2ug sunitinib MP. If so, 140ng (1week post MP injection) and 280ng (2weeks) sunitinib would be available in the eye. However, we have to incorporate 8% loss of free sunitinib per day. If so, 100ng (1week) and 160ng (2weeks) sunitinib would be available in the eye which was injected with 2ug sunitinib MP. These calculations indicated that 500ng free sunitinib injection and 2ug sunitinib MP injection would provide very close amount of free sunitinib 1-2weeks post injection. However, 500ng sunitinib

injection did not suppress laser CNV (Figure 2C), and 2ug sunitinib MP injection suppresses laser CNV (Figure 3B and others). How can you explain this contradiction? Are the microparticles anti-angiogenic in some way?

As shown in Figure 2, empty microparticles do not suppress choroidal neovascularization.

Thirty days after an intravitreal injection of free sunitinib, 8.8% remained in the eye. The reviewer hypothesized that 7 days after injection of 500 ng of free sunitinib, ~280ng would remain in the eye, translating into a half-life of ~ 8.5 days. In making this hypothetical calculation, the reviewer appears to assume a single-compartment pharmacokinetic model is appropriate.

We conducted an additional experiment to test the reviewer's hypothesis. An intravitreal injection of free sunitinib was done in 20 mice and LC/MS was used to measure the amount of sunitinib in 10 eyes immediately after injection at T_0 and at 7 days after injection in 10 eyes. Compared with T_0 , only $10.1 \pm 0.7\%$ of sunitinib remained in the eye 7 days after injection. By fitting these data into the same single-compartment model, the overall half-life of free sunitinib in C57BL/6 mouse eyes is estimated to be ~ 2.1 days, which is similar to the result of a rabbit PK study that we previously conducted (vitreal half-life of ~ 2 days). The results indicate that an intravitreal injection of free sunitinib is cleared from the eye at a much faster rate in the first week than the reviewer estimated. This finding, combined with the sunitinib levels measured at 30 days, is consistent with a two-compartment pharmacokinetic clearance model (rapid clearance phase over the first several days, followed by a slow clearance phase).

The second "compartment" is likely due to the fact that sunitinib has a high binding affinity to melanin. After an intravitreal injection of free sunitinib, some of the drug accumulates in melanin-containing ocular tissues such as the RPE and choroid. Slow release of bound sunitinib from the melanin compartment likely explains the slower clearance rate after the first week. However, as the majority of the unencapsulated sunitinib that remains is bound to melanin within just a few days of injection, it cannot interact with its pharmacological target and suppress CNV. In contrast, also as the reviewer pointed out, the microparticle formulation releases >1% of free sunitinib each day, which serves as a constant source that helps maintain the concentration of free sunitinib (unbound to melanin) in target tissue and, therefore, leads to sustained suppression of CNV.

2. Also, the author should show the rate of cataract by Sunitinib MP injection to prove that MP prevent sunitinib induced cataract (response page 6).

The figure below shows the number of cataracts that were observed after sunitinib MP injection in the experiments for which efficacy data are shown in Figures 2A and B.

Mild cataract was observed in 1 out of 81 eyes that received an intravitreal injection of sunitinib MPs. The mouse that had mild cataract in the sunitinib MP-injected eye also had a similar mild cataract in the fellow eye that was injected with empty MP. Mice occasionally have spontaneous development of mild cataracts, and that was likely the case for this mouse. This information has been added to the manuscript.

In a recently completed GLP repeat-dose ocular toxicity study in minipigs with sunitinib MPs, test article-associated cataract formation was not observed throughout 10 months of the experiment. Therefore, cataract is not a problem after intravitreal injection of sunitinib MPs.

Weiling Yu contributed to new experiment and therefore has been added as an author.

We appreciate the thoughtful comments and suggestions that have helped us to further improve the manuscript.

Sincerely,

Peter A. Campochiaro

REVIEWERS' COMMENTS:

Reviewer #3 (Remarks to the Author):

The authors have made the appropriate modifications to the paper and enhanced the manuscript. I recommend publication.

REVIEWERS' COMMENTS:

Reviewer #3 (Remarks to the Author):

The authors have made the appropriate modifications to the paper and enhanced the manuscript. I recommend publication.

We thank the reviewers for their suggestions that have helped us to improve the manuscript.